# Model-Agnostic Policy Explanations with Large Language Models

**Zhang Xi-Jia**[1]   **Yue Guo**[2]   **Shufei Chen**[1]   **Simon Stepputtis**[2]   **Matthew Gombolay**[1]
**Katia Sycara**[2]   **Joseph Campbell**[3]

[1]Georgia Institute of Technology, Atlanta, GA, USA
{zhang.xijia, schen964}@gatech.edu, matthew.gombolay@cc.gatech.edu

[2]Carnegie Mellon University, Pittsburgh, PA, USA
{yueguo, sstepput, sycara}@andrew.cmu.edu

[3]Purdue University, West Lafayette, IN, USA
joecamp@purdue.edu

## Abstract

Intelligent agents, such as robots, are increasingly deployed in real-world, human-centric environments. To foster appropriate human trust and meet legal and ethical standards, these agents must be able to explain their behavior. However, state-of-the-art agents are typically driven by black-box models like deep neural networks, limiting their interpretability. We propose a method for generating natural language explanations of agent behavior based *only* on observed states and actions – without access to the agent's underlying model. Our approach learns a locally interpretable surrogate model of the agent's behavior from observations, which then guides a large language model to generate plausible explanations with minimal hallucination. Empirical results show that our method produces explanations that are more comprehensible and correct than those from baselines, as judged by both language models and human evaluators. Furthermore, we find that participants in a user study more accurately predicted the agent's future actions when given our explanations, suggesting improved understanding of agent behavior. Importantly, we show that participants are unable to detect hallucinations in explanations, underscoring the need for explainability methods that minimize hallucinations by design.

## 1 Introduction

Rapid advances in artificial intelligence and machine learning have led to an increase in the deployment of robots and other embodied agents in human-centric and safety-critical settings (Sun et al., 2020; Fatima & Pasha, 2017; Li et al., 2023). As such, it is vital that practitioners – who may be laypeople that lack domain expertise or knowledge of machine learning – are able to query such agents for explanations regarding *why* a particular prediction has been made – broadly referred to as explainable AI (Amir et al., 2019; Wells & Bednarz, 2021; Gunning et al., 2019). While progress has been made in this area, prior works tend to focus on explaining agent behavior in terms of rules (Johnson, 1994), vision-based cues (Cruz & Igarashi, 2021; Mishra et al., 2022), semantic concepts (Zabounidis et al., 2023), or trajectories (Guo et al., 2021). However, it has been shown that laypeople benefit from natural language explanations (Mariotti et al., 2020; Alonso et al., 2017) since they do not require specialized knowledge to understand (Wang et al., 2019), leverage human affinity for verbal communication, and increase trust under uncertainty (Gkatzia et al., 2016).

In this work, **we seek to develop a model-agnostic framework to generate natural language explanations of an agent's behavior given only observations of states and actions**. By assuming access to only behavioral observations, we are able to explain behavior produced

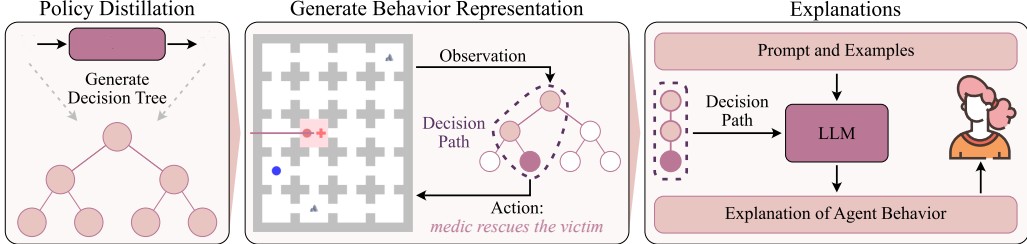

Figure 1: Overview of our three-step pipeline to explain policy actions. Left: a black-box policy is distilled into a decision tree. Middle: a decision path is extracted from the tree for a given state which contains a set of decision rules used to derive the associated action. Right: we utilize an LLM to generate a natural language explanation given the decision path.

by *any* agent policy, including deep neural networks (DNNs). Unlike prior methods that are limited in expressivity due to the utilization of language templates (Hayes & Shah, 2017; Kasenberg et al., 2019; Wang et al., 2019), depend on large corpora of human-written explanations (Ehsan et al., 2019; Liu et al., 2023), or require model fine-tuning (Yang et al., 2025), we propose an approach in which large language models (LLMs) can be used to generate free-form natural language explanations in a few-shot manner. While LLMs have shown considerable zero-shot task performance and are well-suited to generating natural language explanations (Cahlik et al., 2025; Marasović et al., 2021; Li et al., 2022), they are typically applied to commonsense reasoning as opposed to explaining model behavior and are prone to hallucination – a well-known phenomenon in which false information is presented as fact (McKenna et al., 2023). It is an open question as to how LLMs can be conditioned on an agent's behavior in order to generate plausible explanations while avoiding such hallucinations. We find this to be a particularly important aspect, as laypeople tend to struggle to identify hallucinated facts, as we observe in our participant studies in Sec. 5.4.

Our solution, and core algorithmic contribution, is the introduction of a *Behavior Representation* (BR), in which we distill an agent's policy into a locally interpretable model that can be directly injected into a text prompt and reasoned with, without requiring fine-tuning. A behavior representation acts as a compact representation of an agent's behavior around a specific state and indicates what features the agent considers important when making a decision. We show that by constraining an LLM to reason about agent behavior in terms of a behavior representation, we are able to greatly reduce hallucination compared to alternative observation encodings while generating informative and plausible explanations.

Our approach is a three-stage process (see Fig. 1) in which we, 1) distill an agent policy into a decision tree, 2) extract a decision path from the tree for a given state which serves as our local *behavior representation*, and 3) transform the decision path into a textual representation and inject it into pre-trained LLM via in-context learning (Brown et al., 2020) to produce a natural language explanation. Through a series of quantitative experiments and a participant study, we show that a) our approach generates model-agnostic explanations that yields the highest explanation and action prediction accuracy over baseline methods; b) laypeople find our explanations as helpful as those generated by a human domain expert for next action prediction, and prefer our explanations over those generated by baseline methods; and c) our approach yields explanations with significantly fewer hallucinations than alternative methods of encoding agent behavior.

## 2 Related Work

**Explainable Agent Policies**: Many works attempt to explain agent behavior through the use of a simplified but interpretable model that closely mimics the original policy (Puiutta & Veith, 2020; Verma et al., 2018; Liu et al., 2019; Shu et al., 2017), a technique which has long been studied in the field of supervised learning (Ribeiro et al., 2016). Although approaches that directly utilize inherently interpretable models with limited complexity during the training phase (Du et al., 2019) exist, many researchers avoid sacrificing model accuracy

for interpretability. In this work, we follow an approach similar to (Guo et al., 2023), in which we leverage a distilled interpretable model to gain insight into how the agent's policy reasons.

**Natural Language Explanations**: Outside of explaining agent policies, natural language explanations have received considerable attention in natural language processing areas such as commonsense reasoning (Marasović et al., 2020; Rajani et al., 2019) and natural language inference (Prasad et al., 2021). Unlike our setting in which we aim to explain the behavior of a given *model*, these methods produce an explanation purely with respect to the given input and domain knowledge, for example, whether a given premise supports a hypothesis in the case of natural language inference (Camburu et al., 2018). Although self-explaining models (Marasović et al., 2021; Rajagopal et al., 2021; Hu & Clune, 2023) are conceptually similar to our goal, we desire a model-agnostic approach with respect to the agent's policy and thus seek to explain the agent's behavior with a separate model. Approaches that directly prompt LLMs to reason over latent representations of black-box models (Bills et al., 2023) have shown limited success in explaining agent policies, while recent methods like Yang et al. (2025) require additional fine-tuning and reward modeling beyond the agent's state and actions. This motivates our use of an intermediate representation that supports lightweight, generalizable explanation.

## 3 Language Explanations for Agent Behavior

We introduce a framework for generating natural language explanations for an agent from observed state-action pairs. Our approach consists of three steps: 1) we distill the agent's policy into a decision tree, 2) we generate a behavior representation from the decision tree, and 3) we query an LLM for an explanation given the behavior representation. We note that step 1 only needs to be performed once for a particular agent, while steps 2 and 3 are performed each time an explanation is requested. Our method is model-agnostic, making no assumptions about the underlying policy, allowing for explanations to be generated for any model from which trajectories can be sampled.

**Notation**: We consider an infinite-horizon discounted Markov Decision Process (MDP) in which an agent observes environment state $s_t$ at discrete timestep $t$, performs action $a_t$, and receives the next state $s_{t+1}$ and reward $r_{t+1}$ from the environment. The MDP consists of a tuple $(\mathcal{S}, \mathcal{A}, R, T, \gamma)$ where $\mathcal{S}$ is the set of states, $\mathcal{A}$ is the set of agent actions, $R : \mathcal{S} \times \mathcal{S} \to \mathbb{R}$ is the reward function, $T : \mathcal{S} \times \mathcal{A} \times \mathcal{S} \to [0,1]$ is the state transition probability, and $\gamma \in [0,1)$ is the discount factor. As in standard imitation learning settings, we assume the reward function $R$ is unknown and that we only have access to states and actions sampled from a stochastic agent policy $\pi^*(a|s) : \mathcal{A} \times \mathcal{S} \to [0,1]$.

### 3.1 Distilling a Decision Tree

Our first step is to distill the agent's underlying policy into a decision tree, which acts as an interpretable *surrogate*. The decision tree is intended to faithfully replicate the agent's policy while being interpretable, such that we can extract a behavior representation from it. Given an agent policy $\pi^*$, we distill a decision tree policy $\hat{\pi}$ using the DAgger (Ross et al., 2011) imitation learning algorithm, which minimizes the expected loss to the agent's policy under an induced distribution of states,

$$\hat{\pi} = \arg \min_{\pi \in \Pi} \mathbb{E}_{s^*, a^* \sim \pi^*}[\mathcal{L}(s^*, a^*, \pi)], \tag{1}$$

for a restricted policy class $\Pi$ and loss function $\mathcal{L}$. This method performs iterative data aggregation consisting of states sampled from the agent's policy and the distilled decision tree to overcome error accumulation caused by the violation of the i.i.d. assumption. While decision trees are simpler than other methods, e.g. DNNs, it has been shown that they are still capable of learning complex policies (Bastani et al., 2018). Intuitively, DNNs often achieve state-of-the-art performance not because their representational capacity is larger than other models, but because they are easier to regularize and thus train (Ba & Caruana, 2014). Distillation is a technique that can be leveraged to transform the knowledge contained

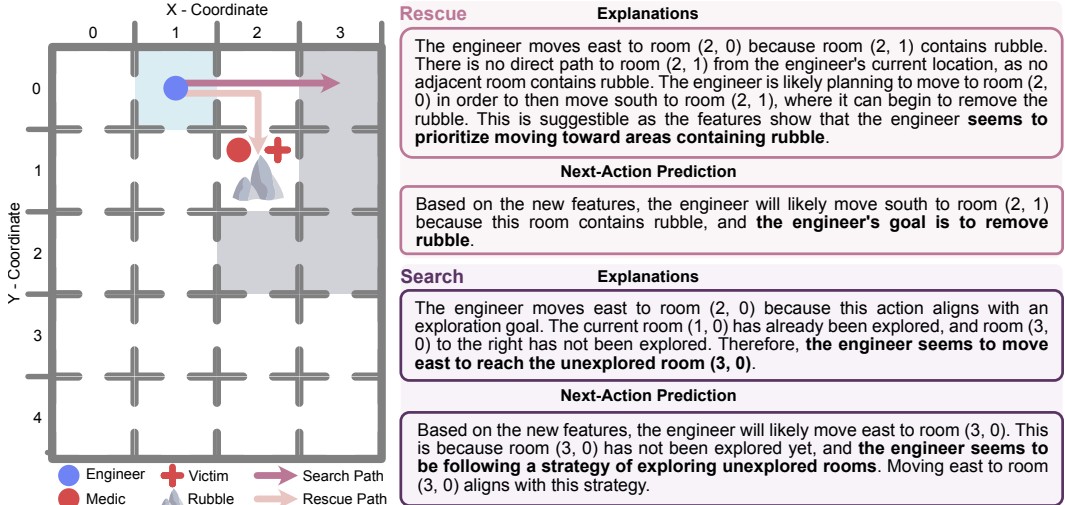

Figure 2: An example of an ambiguous state, in which the engineer's current state can be induced by following two distinct behaviors: Rescue, which prioritizes removing rubble as soon as possible, and Search, which prioritizes visiting unexplored rooms. Given the current state (engineer at (1, 0)) and intended action (going east), their follow-up action is ambiguous depending on which behavior is utilized by the engineer (Search: Purple; Rescue: Pink). The corresponding decision paths are shown for each possible behavior and the resulting natural language explanations after transforming into a behavior representation.

within a DNN into a more interpretable decision tree (Hinton et al., 2015; Frosst & Hinton, 2017; Bastani et al., 2017).

## 3.2 Behavior Representation Generation

The distilled policy $\hat{\pi}$ consists of a set of decision rules that approximate the decision-making process of the agent's policy $\pi^*$. Given a state $s_t$ and action $a_t$ taken by the agent, we extract a decision path $dp = Path(\hat{\pi}, s_t)$ which acts as a *locally* interpretable model of the agent's behavior. The path $dp$ consists of a subset of the decision rules in $\hat{\pi}$ which produce the action $a_t$ in state $s_t$, and is obtained by simply traversing the tree from root to leaf. While a decision tree does *not* explicitly model causality, the decision rules approximate the agent's underlying decision-making rationale in state $s_t$ and can be used to **infer possible intent**.

Figure 2 shows example decision paths for agents operating in an Urban Search and Rescue (USAR) environment where heterogeneous agents with different action spaces learn to coordinate to rescue victims (Lewis et al., 2019; Freeman et al., 2021). USAR is a multi-agent, partially observable environment with two agents: a *Medic*, responsible for healing victims and an *Engineer* responsible for clearing rubble. The top decision path, denoted as Rescue, corresponds to an agent exhibiting behavior that prioritizes removing rubble as it is discovered. The bottom decision path, Search, corresponds to behavior prioritizing exploration, i.e., it fully explores the environment before removing any pieces of rubble. We can observe how these different behaviors are reflected in their respective decision paths – the Search path largely consists of decision rules examining whether rooms have been explored, while the Rescue path consists of rules checking for the existence of rubble. This enables reasoning, e.g., since the Search agent *only* checks for unexplored rooms before taking its action, we can infer that the agent is currently interested in exploration as it is choosing to ignore visible rubble.

We refer to such a decision path as a behavior representation, and it serves as a compact encoding of the agent's behavior. This representation is effective for three reasons: a) decision paths provide an intuitive and explicit ordered set of rules with which an LLM can reason, resulting in more effective explanations and reduced hallucination compared to alternative behavior encodings; b) decision paths can be readily translated into natural

| Top-1 Rank ↑ | | USAR RESCUE | | | | | USAR SEARCH | | | | | BABYAI | | | PACMAN | | |
|---|---|---|---|---|---|---|---|---|---|---|---|---|---|---|---|---|---|
| Model | Method | Cor. | Inf. | Str. | Cat. | Goal | Cor. | Inf. | Str. | Cat. | Goal | Cor. | Inf. | Goal | Cor. | Inf. | Goal |
| GPT-4 | **Path (Ours)** | **0.50** | **0.42** | **0.53** | **0.59** | **0.41** | **0.46** | **0.45** | **0.52** | **0.59** | **0.51** | 0.43 | 0.45 | **0.47** | 0.32 | **0.49** | **0.46** |
| | State | 0.29 | 0.28 | 0.32 | 0.21 | 0.31 | 0.26 | 0.32 | 0.35 | 0.28 | 0.30 | 0.23 | 0.29 | 0.27 | 0.28 | 0.29 | 0.22 |
| | No BR | 0.21 | 0.30 | 0.15 | 0.20 | 0.28 | 0.28 | 0.23 | 0.13 | 0.14 | 0.18 | 0.34 | 0.26 | 0.26 | **0.40** | 0.22 | 0.32 |
| GPT-3.5 | **Path (Ours)** | **0.48** | **0.53** | **0.54** | **0.57** | **0.60** | **0.46** | **0.50** | **0.54** | **0.57** | **0.49** | **0.80** | **0.75** | **0.75** | 0.37 | **0.43** | **0.39** |
| | State | 0.18 | 0.15 | 0.20 | 0.17 | 0.14 | 0.27 | 0.28 | 0.32 | 0.27 | 0.30 | 0.10 | 0.12 | 0.13 | 0.32 | 0.31 | 0.31 |
| | No BR | 0.34 | 0.32 | 0.26 | 0.26 | 0.27 | 0.28 | 0.22 | 0.14 | 0.17 | 0.21 | 0.10 | 0.13 | 0.12 | 0.31 | 0.26 | 0.30 |
| Llama-2 | **Path (Ours)** | **0.46** | **0.50** | **0.54** | **0.56** | **0.52** | **0.50** | **0.43** | **0.51** | **0.56** | **0.45** | **0.54** | **0.52** | **0.55** | **0.57** | **0.57** | **0.59** |
| | State | 0.22 | 0.16 | 0.22 | 0.19 | 0.17 | 0.25 | 0.29 | 0.28 | 0.24 | 0.23 | 0.14 | 0.14 | 0.15 | 0.17 | 0.15 | 0.15 |
| | No BR | 0.32 | 0.34 | 0.24 | 0.25 | 0.31 | 0.26 | 0.28 | 0.21 | 0.20 | 0.32 | 0.32 | 0.34 | 0.30 | 0.26 | 0.28 | 0.26 |
| Starling | **Path (Ours)** | **0.68** | **0.67** | **0.68** | **0.71** | **0.64** | **0.71** | **0.63** | **0.77** | **0.78** | **0.69** | **0.55** | **0.51** | **0.60** | 0.38 | **0.45** | **0.41** |
| | State | 0.17 | 0.17 | 0.22 | 0.18 | 0.23 | 0.17 | 0.21 | 0.11 | 0.13 | 0.17 | 0.20 | 0.24 | 0.20 | 0.28 | 0.31 | 0.27 |
| | No BR | 0.15 | 0.16 | 0.10 | 0.11 | 0.13 | 0.13 | 0.16 | 0.12 | 0.09 | 0.14 | 0.25 | 0.25 | 0.20 | 0.34 | 0.24 | 0.32 |

Table 1: Ratio in which the explanation generated by each method is ranked as best according to a given set of criteria over 100 samples. Explanations are generated with multiple language models over different environments. Rankings are automatically generated by an LLM according to the criteria detailed in Sec. 4.1. Values are bolded if they are significantly better than all other methods for a given model according to a paired $t$-test ($p$-value $< 0.05$).

language via algorithmic templates and injected into an LLM prompt – meaning no fine-tuning is required (Brown et al., 2020), which is an important factor given the lack of human-annotated explanations for agent behavior; and c) even deep decision trees representing complex policies can fit into relatively small LLM context windows.

### 3.3 In-Context Learning with Behaviors

The last step in our approach is to define a prompt that constrains the LLM to reason about agent behavior with respect to a given behavior representation. Our prompt consists of four parts: a) a concise description of the environment the agent is operating in, e.g., state and action descriptions, b) a description of what information the behavior representation conveys, c) in-context learning examples, and d) the behavior representation and action that we wish to explain. An example of this prompt is shown in the appendix. All parts except for (d) are pre-defined ahead of time and remain constant for all queries, while our framework provides a mechanism for automatically constructing (d). Thus, our system can be queried for explanations with no input required by the user.

## 4 Quantitative Results and Analysis

We quantitatively evaluate the performance of our proposed approach in three different environments with the goal of answering the following questions: **1)** Does our approach enable the LLM to reason about observed agent behavior in order to produce informative and correct explanations? **2)** Does our approach enable the LLM to infer *future* behavior? **3)** How is hallucination affected by our choice of behavior representation?

**Tasks and Environments**: We evaluate our method with agent behaviors drawn from three different environments. **Urban Search and Rescue (USAR)** is a partially observable multi-agent cooperative task requiring the agents to navigate a 2D Gridworld environment to rescue victims and remove rubble. The agents have a heterogeneous set of actions. The engineer can remove rubble that may be hiding victims, and the medic can rescue victims after the rubble has been removed. The agents may act according to three types of behaviors: Search in which the agents prioritize exploration over removing rubble and rescuing victims; Rescue in which the agents prioritize removing rubble and rescuing victims, and Fixed in which the agents ignore rubble and victims and perform a fixed navigation pattern. **Pacman** is based on the classic video game in which the agent attempts to eat food pellets while avoiding being eaten by ghosts. If the agent consumes a power pellet, the ghosts may, in

| Accuracy ↑ | | EXPLANATION | | | | | | | | ACTION PREDICTION | | | | | |
| | | Long-term | | | Short-term | | | Ambiguous | | | Long-term | | Short-term | | Ambiguous | |
| Env. | Method | Str. | Cat. | Goal | Str. | Cat. | Goal | Str. | Cat. | Goal | Act. | Int. | Act. | Int. | Act. | Int. |
| USAR Rescue | **Path (Ours)** | 0.70 | 0.75 | **0.75** | **1.00** | **1.00** | **1.00** | **0.90** | **0.85** | **0.85** | **0.80** | **0.75** | 0.40 | **0.75** | **0.85** | **0.85** |
| | State | **0.75** | **0.80** | **0.75** | 0.75 | 0.75 | 0.75 | 0.60 | 0.75 | 0.75 | 0.65 | 0.55 | **0.75** | **0.75** | 0.80 | 0.70 |
| | No BR | 0.25 | 0.25 | 0.25 | 0.90 | 0.95 | 0.95 | 0.70 | 0.75 | 0.75 | 0.55 | 0.50 | 0.65 | 0.65 | 0.80 | 0.75 |
| USAR Search | **Path (Ours)** | **0.90** | **0.75** | **0.25** | **1.00** | **1.00** | 0.70 | **0.90** | **0.80** | **0.35** | **0.95** | **1.00** | **0.80** | **0.95** | **0.90** | **0.95** |
| | State | 0.40 | 0.05 | 0.05 | 0.70 | 0.95 | **0.95** | 0.00 | 0.00 | 0.00 | 0.60 | 0.40 | 0.45 | 0.45 | 0.05 | 0.05 |
| | No BR | 0.20 | 0.30 | 0.15 | 0.40 | 0.90 | 0.90 | 0.00 | 0.05 | 0.00 | 0.65 | 0.25 | 0.50 | 0.35 | 0.30 | 0.10 |

Table 2: (Left) Explanation accuracy for randomly sampled states in USAR in which the agent is pursuing a long-term goal, short-term goal, or ambiguous goal while operating under two different behavior strategies: Search and Rescue. All metrics represent accuracy (higher is better). Each value is computed by-hand by a human domain expert over 20 samples: 10 each from medic and engineer. The best method in each column is bolded. (Right) Action prediction accuracy for randomly sampled states in which the agent is pursuing a long-term goal, short-term goal, or ambiguous goal. Action (Act.) indicates whether the next action was correctly identified, while Intent (Int.) indicates whether a possible reason for *why* the action was taken was correctly identified.

turn, be eaten for a limited time for a higher reward. **BabyAI** (Hui et al., 2020) is a set of 2D Gridworld environments in which an agent is required to solve language-conditioned tasks. In this work, we sample agent behaviors from the Gotolocal environment in which an agent must navigate to a goal object amid distractors.

**Models**: In order to evaluate the effect our method has on various large language models, we generate explanations using four popular models: GPT-4 (OpenAI, 2023), GPT-3.5 (Brown et al., 2020), Starling (Zhu et al., 2024), and LLama 2 (Touvron et al., 2023).

**Methods**: We generate natural language explanations using one of three methods: **Path** is our proposed method, which uses a decision path as a behavior representation, **State** is an alternative behavior representation that uses sequences of state-action pairs sampled from the agent's policy rather than a decision path, and **No BR** is an ablated baseline in which no behavior representation is given. Further details on all environments, models, and methods can be found in Appendix A.4.

## 4.1 Evaluating Comprehensibility and Accuracy

We first evaluate how our approach affects the quality of generated explanations. Unlike works that evaluate natural language explanations in domains such as natural language inference, to the best of our knowledge, there are no datasets consisting of high-quality explanations generated over agent behavior. Subsequently, we create annotated datasets by sampling agent behaviors from heuristic policies in each environment. Using a heuristic policy allows us to access the internal beliefs and intents of the agent and programmatically generate ground truth explanations. We generate explanations for each observation in this dataset using the methods described previously and automatically rank them using an LLM (GPT-4o) (Zhu et al., 2024) according to five criteria: **informativeness** which measures whether an explanation contains sufficient information to understand an action; **correctness** which measures whether an explanation contains accurate information; and whether an explanation identifies an agent's **strategy**, **goal**, and goal **category**. Definitions for each criteria can be found in Appendix A.3. Our results are shown in Table 1 where we observe the following.

**Explanations produced with Path are ranked highest across all environments and models.** In particular, we note that our method produces better rankings in Llama-2 and Starling, suggesting that the decision path representation is especially useful in smaller, resource-constrained models.

## 4.2 Evaluating Explanation Quality

Next, we manually analyze GPT-4-generated explanations in the USAR environment to quantify explanation accuracy and hallucination rate more accurately. States are grouped into three categories: **Long-term** — the agent is moving to a room/rubble/victim but won't get there in the next time step; **Short-term** — the agent is pursuing a short-term goal, meaning it will reach the desired room/remove rubble/rescue victim in the next time step; and **Ambiguous** — the *current* state-action can be induced by either searching or rescuing behaviors, but the *next* state will yield different actions from each behavior. The results for each behavior are shown in Table 2 and Fixed in Appendix A.2.2. We make the following observations.

**Explanations produced with Path are more accurate**. Explanations generated using a decision path behavior representation more accurately identify the agent's Strategy, Category, and Goal in every category except for long-term Rescue when compared to other methods. We conjecture that the reduced accuracy for long-term goals under Rescue is due to the additional complexity associated with the decision paths; they must simultaneously check for unexplored rooms and rubble, while the Search decision paths do this sequentially, i.e., they first check all rooms' exploration status and *then* check for the presence of rubble.

**LLMs make assumptions over expected behaviors**. The ambiguous states reveal an important insight: the LLM (here GPT-4) tends to assume an agent will act as Rescue when presented with an observation and a task description. We can see that all methods, including State and No BR, yield relatively high accuracy when generating explanations for ambiguous state-actions under Rescue behavior. However, when the agent acts under a Search policy, Path continues to perform well (90% accuracy) while the other methods fail to get *any* explanations correct (0% accuracy). We find that this is because when presented with an ambiguous state that could be caused by multiple behaviors, the LLM assumes the agent acts as Rescue and only the decision path behavior representation is able to enforce a strong enough prior over agent behavior for a correct explanation. A similar trend can be observed with states sampled from the Fixed behavior in Table 5. Path yields an impressive 80% accuracy in detecting that the agent is ignoring victims and rubble and pursuing a pre-determined path. Yet again, the LLM assumes Rescue behavior for State and No BR and yields nearly no correct explanations.

We further evaluate how well the LLM is actually able to reason over and explain current agent behavior by analyzing how well it can predict *future* agent behavior. Intuitively, if the LLM can accurately produce an explanation that identifies an agent's intent, then it should be able to use this intent to infer future actions. We evaluate this by issuing a follow-up prompt to the LLM for each explanation produced in the previous analysis to predict the agent's next action given its previously generated explanation.

**Explanations produced with Path enable accurate action prediction.** Tables 2 and 5 show that the LLM is able to effectively predict the agent's next actions when reasoning with explanations produced by Path, consistently yielding 80-90% accuracy across all behavior types. There is one exception to this: short-term goal action prediction, which yields 40% accuracy. This is due to the *locality* of the decision path – the path only encodes decision rules relevant to the agent's current action, which is highly correlated with the agent's current goal. If that goal happens to be short-term, meaning it will be achieved with the agent's *current* action, then the decision path rarely encodes enough information to reason about the *next* action. We conjecture that providing additional information, such as the current observation, in addition to the decision path behavior representation, can alleviate this issue. This is also the cause for the low action prediction accuracy over Fixed states in Table 5; the agent's strategy is often identified but there is not enough information to predict the next action.

### 4.3 Evaluating hallucination

Fig. 3 shows hallucination frequencies in explanations and action predictions.

**Hallucination is significantly reduced with Path.** There are two key insights from our hallucination evaluation: a) decision path-based explanations yield far lower hallucination rates across all categories, and b) State sometimes yields *more* hallucinations than No BR. This may initially appear to be a counterintuitive result. However, upon inspection, evidence suggests that when no behavior representation is provided to the LLM, the LLM tends to be cautious in making assumptions, resulting in fewer hallucinations at the cost of lower explanation and action prediction accuracy.

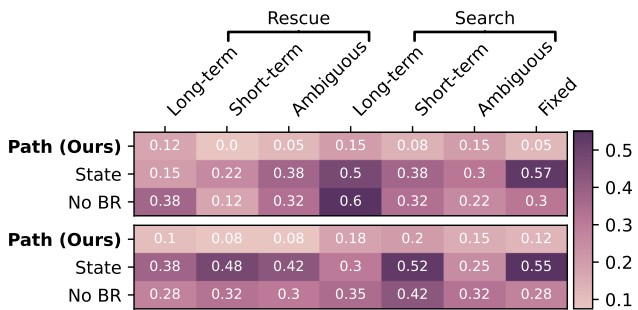

Figure 3: Hallucination rates in generated explanations (top) and action predictions (bottom).

**Hallucination is not correlated with action prediction accuracy.** Intuitively, we might think that the hallucination rate of the generated explanations is inversely correlated with the action prediction accuracy. That is, hallucinations are symptomatic of LLM uncertainty regarding agent intent and inject additional errors into the downstream reasoning process. However, we find this is not the case with no significant correlations between hallucination and action prediction metrics according to the Pearson correlation coefficient with $p < 0.05$.

## 5 Participant Study and Analysis

While the above analysis shows that our method produces accurate explanations, we seek to answer whether the explanations are actually *helpful* to humans. We evaluate helpfulness by examining whether humans prefer our explanations and whether those explanations enhance their understanding of the agent's behavior, as measured by improved performance in predicting the agent's next action. We conduct an IRB-approved, within-subject participant study where 29 recruited participants are given a set of randomized questions from a randomly selected set of cases from the USAR environment. Additional details can be found in Appendix A.6. The study includes two tasks: (1) predicting the AI agent's next move, first *without* and then *with* explanations from each method, and (2) comparing two explanations pairwise to rank their helpfulness. In both tasks, participants are also asked to identify whether the explanations contain any hallucinations.

The following hypotheses guide our evaluation. Here, Path is our proposed method and State is an alternative baseline as defined in Sec. 4; Template is a textual representation of the decision path; and Human refers to explanations produced by a human domain expert. Human is intended to serve as an upper-bound on explanation quality, and Template a lower-bound.

**H1:** Participants predict the next action more accurately with explanations generated by Path compared to State (**H1.1**), Template (**H1.2**) and no explanation at all (**H1.3**), with no significant difference in accuracy compared to Human (**H1.4**).

**H2:** Participants prefer explanations generated by Path compared to State (**H2.1**) and Template (**H2.2**), with no significant difference in preference compared to Human (**H2.3**).

**H3:** Participants predict the next action less accurately when using explanations that contain hallucinations – both subjectively perceived (**H3.1**) and objectively present (**H3.2**) – compared to when explanations are hallucination-free.

**H4:** Explanations containing hallucinations – both subjectively perceived (**H4.1**) and objectively present (**H4.2**) – receive lower preference ratings than those without perceived or real hallucinations.

## 5.1 Evaluating Helpfulness through Action Prediction Performance (H1)

To test **H1** (and **H3**), participants are presented with a trajectory of an agent and are completely unaware of the behavior pattern the agent might follow. They are first asked to predict the agent's next action. Then, participants are given an explanation generated by Path, State, Template, or Human, asked to label any factually incorrect information or illogical reasoning, and then asked to predict the agent's next action again.

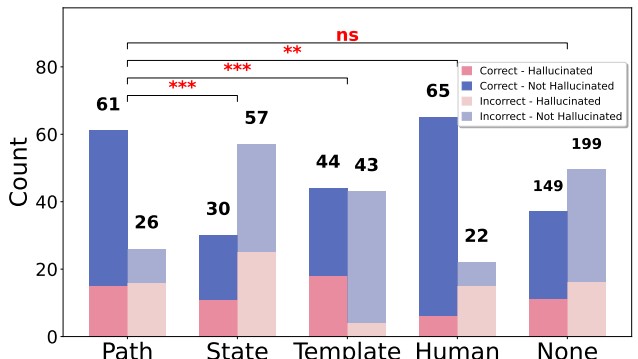

Using binary logistic regression with controls, we find that **H1.1** ($coef = +2.650$, $p < 0.001$), **H1.2** ($coef = +2.898$, $p < 0.001$), and **H1.3** ($coef = +1.390$, $p < 0.01$) are all supported with significance (see Fig. 4). That is, participants are significantly more accurate in predicting the agent's next action when provided with explanations generated by our method, compared to those from State, Template, or no explanation at all. A McNemar's test shows no significant difference in accuracy between Path and Human explanations, supporting the hypothesis **H1.4** ($p > 0.05$) that our explanations perform on par with those written by human experts.

Figure 4: Number of correct and incorrect predictions under different methods. None refers to the no-explanation baseline and is downsampled by a factor of 4 for comparability. *** indicates $p < 0.001$ (statistically significant); ** indicates $p < 0.01$ (statistically significant); ns: not significant. Hypotheses that are supported are labeled in red.

## 5.2 Evaluating Helpfulness through Participant Preference (H2)

To test **H2** (and **H4**), participants are given three types of pairwise comparisons, Path vs State, Path vs Template, and Path vs Human. All pairs are presented as Option A and Option B with randomized orders. Participants are asked to label any factually incorrect information or illogical reasoning, and then indicate their preference between Option A and Option B on a five-point Likert response scale, reflecting how helpful the explanation was in understanding the agent's next action.

Using Wilcoxon signed-rank tests, we find that **H2.2** ($W = 3016.00$, $p < 0.001$) is supported with significance: participants show a clear preference for explanations generated by our method over those from Template. In contrast, **H2.1** ($W = 1778.00$, $p > 0.05$) is not supported, indicating no significant difference in preference between our method and State. **H2.3** ($W = 741.00$, $p < 0.001$) is also not supported, with participants significantly preferring Human explanations over those generated by our method. See Fig. 5 for full results.

In follow-up interviews, some participants acknowledged the State explanations did not align with the current agent's strategy and yet still preferred them more or equally. They justified this by suggesting that such explanations could still be valid under other hypothetical policies. This suggests that humans tend to be overly generous in accepting explanations, which is a potential confounder.

## 5.3 Evaluating Hallucination (H3, H4)

Consistent with LLM-based action prediction results in Sec. 4.1, we find that participant action prediction accuracy is not significantly affected by perceived hallucination. Specifically, **H3.1** is only supported in Path vs Human comparisons ($coef = -1.713$, $p < 0.001$) while **H3.2** ($p > 0.05$) is not supported. Regarding participant preferences, we observe that **H4.1** and **H4.2** are only supported in the Path-vs.-Template comparison ($coef = -1.763$, $p = 0.01$ and $coef = -1.049$, $p < 0.05$), indicating that hallucination does not significantly influence pref-

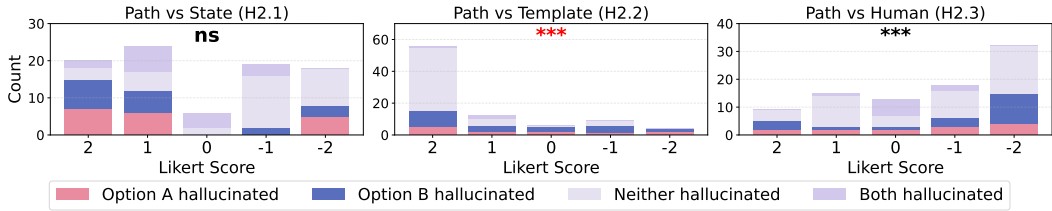

Figure 5: 5-point Likert response scale describing participants' preferences of option A vs. option B under different hallucination conditions. *** indicates $p < 0.001$ (statistically significant); ns: not significant. Hypotheses that are supported are labeled in red.

erence in general. Interestingly, subjectively perceived hallucinations in `Template` strongly decreases preference for `Path` ($coef = -1.758, p < 0.01$).

To test whether this phenomenon is tied to participant's inability to detect hallucinations, we further perform another binary logistic regression with controls using hallucination labels from both tasks. Results suggest that there is no significance ($coef = +0.482, p > 0.05$) between objectively present and subjectively perceived hallucination, indicating participants were unable to detect hallucinated information.

### 5.4  Key Takeaways

**Explanations produced with `Path` enable accurate action prediction.** Consistent with the results of LLM-based action prediction (Sec. 4.2), this suggests that our proposed approach generates explanations which yield an improved understanding of agent behavior *for both humans and LLMs*.

**Hallucination – both objectively present and subjectively perceived – generally are neither correlated with action prediction accuracy nor helpfulness.** This indicates that either a) hallucinated facts do not diminish explanation helpfulness in the eyes of participants, as supported by follow-up interviews, or b) participants fail to identify hallucinations, as supported by hallucination evaluation.

**Humans cannot reliably detect hallucinations.** This suggests that we should not depend on post-hoc detection by users – we need to prevent hallucinations at the source.

## 6  Conclusion and Future Work

In this work, we propose a model-agnostic framework for producing natural language explanations for an agent's behavior. Through the construction of a *behavior representation*, we are able to prompt an LLM to reason about agent behavior in a way that produces plausible and useful explanations while limiting hallucinations, as measured through a participant study and empirical experiments. While we recognize that our proposed method has limitations, namely that it requires distilling an agent's policy into a decision tree that only works with non-dense inputs, we believe this represents a valuable path toward developing explainable policies. Such limitations can be overcome with more complex behavior representations, e.g., differentiable decision trees or concept feature extractors, and we expect the quality of explanations to improve as LLMs improve.

## 7  Acknowledgment

This work has been funded by DARPA ANSR: FA8750-23-2-1015, ARL-W911QX24F0049, and the National Science Foundation award IIS-2340177.

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

# A  Appendix

## A.1  Limitations

Our work currently has several limiting assumptions. First, we assume that the task for which explanations are generated falls within the pre-trained LLM's training data distribution. That is, we don't expect this approach to work if it requires specialized knowledge, e.g. neurosurgical expertise, if that is lacking from the LLM's training data. Along these lines, we conjecture that our approach is better-suited for high-level explanations, e.g. the robot went to the kitchen, rather than low-level granular explanations, e.g. the robot applied a trajectory of velocity commands. Second, we currently leverage a decision tree as a behavior representation, which requires structured inputs and precludes operating over dense, unstructured inputs such as images. However, we believe this limitation can be overcome by employing alternative models for behavior representations, e.g. feature extractors or differentiable decision trees. Lastly, as with all post-hoc explainability methods, there is no guarantee that the generated explanations represent the *actual* decision making process of the underlying agent model. However, in practice an explanation only needs to be faithful enough to be *useful*, which in turn depends on the purpose and audience. For example, humans ascribe mental models to explain the behavior of other humans, pets, or even robots so that we may plan our own actions. These models may not be fully accurate to the underlying reasoning process, but if they roughly capture high-level behavior then they are still useful. We view post-hoc explanations such as the ones presented in this work in the same light.

In addition, our participant study has two notable limitations. First, our sample population of university students may not be representative of potential users. Second, for participant helpfulness ratings, we used a single Likert item for comparisons—this isn't ideal and a Likert scale would generally be better. However, we opted for a Likert item because we felt that administering a full Likert scale would result in survey fatigue.

## A.2  Additional Results and Analysis

### A.2.1  *Prompt Sensitivity Analysis*

Prior works have shown that LLMs are often sensitive to the specific wording and formatting of textual prompts (Sclar et al., 2023). To investigate the sensitivity of our generated explanations to the prompt format, we perform experiments in which we apply two of the alterations from the FORMATSPREAD algorithm (Sclar et al., 2023): task 280 and task 904. After modifying the prompt, we generate new explanations using the Path behavior representation and re-run the LLM-based rankings from Table 1. Results in Table 3 and 4 show minimal differences compared to those with the original prompts in Table 1. Explanations produced with Path are consistently ranked highest for most settings. While there is a minor ranking degradation, it is minimal and suggests that our approach is fairly insensitive to prompt format.

Table 5 shows results from the manual analysis of explanation and action prediction accuracy for the Fixed behavior, as in Sec. 4.2. From this, we make the following additional observation.

**Predictions can be right for the wrong reasons.** The State and No BR methods perform worse in action prediction accuracy and approximately align with explanation accuracy, indicating that in most cases, it is difficult to predict future behavior if the agent's decision-making rationale cannot be identified. However, there is an exception to this, which is the relatively high accuracy of 60% for State when predicting over Fixed policy states (Table 5). On analysis, we found that the LLM can identify the simple action distribution produced by the pre-determined path (the agent moves in a north-south pattern) from the set of provided state-action samples, which is further narrowed down by spatial reasoning constraints, e.g., the agent can't move further north if it is already in the northern-most row. However, the LLM is unable to reason about *why* the agent follows such an action distribution, leading to a case where actions are predicted correctly but the agent's rationale is not.

| Top-1 Rank ↑ | | USAR RESCUE - 280 | | | | | USAR SEARCH - 280 | | | | |
|---|---|---|---|---|---|---|---|---|---|---|---|
| Model | Method | Cor. | Inf. | Str. | Cat. | Goal | Cor. | Inf. | Str. | Cat. | Goal |
| GPT-4 | **Path** | **0.52** | **0.58** | **0.65** | **0.64** | **0.56** | **0.52** | **0.57** | **0.62** | **0.64** | **0.62** |
| | State | 0.24 | 0.21 | 0.25 | 0.18 | 0.22 | 0.27 | 0.28 | 0.19 | 0.17 | 0.18 |
| | No BR | 0.24 | 0.21 | 0.10 | 0.17 | 0.22 | 0.21 | 0.15 | 0.18 | 0.18 | 0.19 |
| GPT-3.5 | **Path** | **0.51** | **0.56** | **0.58** | **0.61** | **0.65** | **0.55** | **0.57** | **0.62** | **0.58** | **0.62** |
| | State | 0.22 | 0.15 | 0.17 | 0.18 | 0.17 | 0.27 | 0.2 | 0.17 | 0.18 | 0.18 |
| | No BR | 0.27 | 0.29 | 0.25 | 0.21 | 0.17 | 0.18 | 0.23 | 0.21 | 0.24 | 0.19 |
| Llama-2 | **Path** | 0.32 | 0.37 | **0.46** | **0.51** | **0.49** | 0.29 | 0.31 | **0.49** | **0.62** | 0.45 |
| | State | 0.31 | 0.28 | 0.22 | 0.19 | 0.22 | 0.34 | 0.29 | 0.28 | 0.21 | 0.28 |
| | No BR | 0.37 | 0.36 | 0.32 | 0.29 | 0.29 | 0.37 | 0.39 | 0.23 | 0.17 | 0.27 |
| Starling | **Path** | **0.64** | **0.65** | **0.71** | **0.71** | **0.68** | **0.62** | **0.67** | **0.78** | **0.78** | **0.7** |
| | State | 0.22 | 0.21 | 0.16 | 0.18 | 0.16 | 0.22 | 0.22 | 0.12 | 0.12 | 0.21 |
| | No BR | 0.14 | 0.14 | 0.14 | 0.11 | 0.17 | 0.16 | 0.11 | 0.10 | 0.10 | 0.09 |

Table 3: Ratio in which the explanation generated by each method is ranked as best according to a given set of criteria over 100 samples. Explanations are generated with altered prompts adhereing to task id 280 (Sclar et al., 2023). Rankings are automatically generated by an LLM according to the criteria detailed in Sec. 4.1. Values are bolded if they are significantly better than all other methods for a given model according to a paired $t$-test ($p$-value $< 0.05$).

| Top-1 Rank ↑ | | USAR RESCUE - 904 | | | | | USAR SEARCH - 904 | | | | |
|---|---|---|---|---|---|---|---|---|---|---|---|
| Model | Method | Cor. | Inf. | Str. | Cat. | Goal | Cor. | Inf. | Str. | Cat. | Goal |
| GPT-4 | **Path** | **0.57** | **0.62** | **0.63** | **0.69** | **0.61** | **0.50** | **0.58** | **0.61** | **0.63** | **0.63** |
| | State | 0.27 | 0.22 | 0.22 | 0.16 | 0.20 | 0.32 | 0.24 | 0.20 | 0.18 | 0.18 |
| | No BR | 0.17 | 0.16 | 0.15 | 0.16 | 0.19 | 0.18 | 0.18 | 0.19 | 0.18 | 0.18 |
| GPT-3.5 | **Path** | **0.57** | **0.60** | **0.61** | **0.62** | **0.63** | **0.63** | **0.64** | **0.68** | **0.69** | **0.65** |
| | State | 0.19 | 0.15 | 0.2 | 0.16 | 0.19 | 0.20 | 0.18 | 0.15 | 0.15 | 0.18 |
| | No BR | 0.24 | 0.26 | 0.18 | 0.22 | 0.17 | 0.17 | 0.17 | 0.17 | 0.17 | 0.17 |
| Llama-2 | **Path** | **0.53** | **0.58** | **0.61** | **0.62** | **0.67** | 0.44 | **0.53** | **0.61** | **0.64** | **0.62** |
| | State | 0.21 | 0.17 | 0.17 | 0.16 | 0.17 | 0.29 | 0.25 | 0.21 | 0.17 | 0.23 |
| | No BR | 0.26 | 0.26 | 0.23 | 0.22 | 0.16 | 0.27 | 0.22 | 0.18 | 0.19 | 0.15 |
| Starling | **Path** | **0.61** | **0.67** | **0.71** | **0.72** | **0.72** | **0.67** | **0.68** | **0.77** | **0.81** | **0.72** |
| | State | 0.24 | 0.2 | 0.15 | 0.13 | 0.17 | 0.24 | 0.26 | 0.13 | 0.09 | 0.20 |
| | No BR | 0.16 | 0.13 | 0.15 | 0.16 | 0.11 | 0.09 | 0.06 | 0.10 | 0.10 | 0.08 |

Table 4: Ratio in which the explanation generated by each method is ranked as best according to a given set of criteria over 100 samples. Explanations are generated with altered prompts adhereing to task id 904 (Sclar et al., 2023). Rankings are automatically generated by an LLM according to the criteria detailed in Sec. 4.1. Values are bolded if they are significantly better than all other methods for a given model according to a paired $t$-test ($p$-value $< 0.05$).

### A.2.2  Explanation Quality for USAR Fixed

| | Method | Strategy | Action | Intent |
|---|---|---|---|---|
| | | Accuracy ↑ | | |
| Fixed | **Path (Ours)** | **0.80** | 0.40 | **0.25** |
| | State | 0.05 | **0.65** | 0.00 |
| | No BR | 0.00 | 0.35 | 0.00 |

Table 5: Explanation and action prediction accuracy values for the `Fixed` policy. Since there is no goal or category, only Strategy is shown.

### A.2.3  Human Preference Alignment

To further understand how well human preferences align with explanation correctness and utility, we conducted additional statistical analyses on our participant data. In examining how preferences align with correctness, we found that participants tend to be overly generous in accepting explanations that did not reflect the current agent's strategy (as discussed in Section 5.2), and instead appear to rely on other factors when making their decisions. To investigate what these factors might be, we ran a logistic regression to assess whether explanation length influenced preference ratings. We found that longer explanations were significantly less likely to be preferred ($p < 0.001$ without controls; marginally significant at $p \approx 0.053$ with controls). This suggests that the relationship between preference and correctness may be confounded or even dominated by differences in explanation length. In practice, we observed that participants tended to select shorter responses, regardless of their fidelity.

As for explanation utility, we note that presenting both explanations side by side made it difficult to isolate the attention participants gave to each. However, logistic regression on total time spent per question found no evidence that longer engagement led to better preference for non-hallucinated explanations ($p = 0.43$ without controls; $p = 0.38$ with controls), nor to more accurate hallucination detection ($p = 0.74$ without controls; $p = 1.00$ with controls). This supports our finding that participants were not able to reliably detect hallucinations (Sec. 5.3), and further motivates the need for developing methods that inherently produce correct explanations.

## A.3  LLM Ranking Methodology

Following the procedure laid out in Zhu et al. (2024), we prompt an LLM – specifically GPT-4o – to rank explanations produced by each method from best to worst according to a given set of criteria. These rankings serve as a cost-effective method of evaluating the relative quality of explanations without requiring human supervision. Our procedure is as follows:

1. For each triplet of explanations (one for each method) produced for a single state-action pair, for a single language model, for a single environment, we prompt the LLM to rank the explanations from best to worst.

2. We provide a single criteria to rank the explanations along with a programmatically generated **ground truth explanation** containing objective facts.

3. We then present each explanation in a **random order** so as to avoid order bias.

4. The output is made to conform to a TypeScript Microsoft (2023) schema which is then automatically converted into a ranking statistic that determines which explanation was ranked as "best".

The prompt is as follows:

You are an AI agent which is ranking the quality of explanations produced by different machine learning models. These explanations attempt to describe why another AI agent has taken a particular action. Your task is to compare the explanations to each other to determine the relative ordering from best to worst according to a given criteria and a ground truth explanation which provides objective facts.

Criteria: [criteria definition]

Ground truth: [ground truth explanation generated via template]

Explanations:

1: [random explanation 1]

2: [random explanation 2]

3: [random explanation 3]

### A.3.1 Criteria Definitions

Loosely inspired by the aspects defined in Fu et al. (2023), we define the following criteria.

Task-Agnostic Criteria:

- Correctness: "Rank the explanations from best to worst according solely to their faithfulness to the facts in the given ground truth explanation. That is, any factual information should be re-stated in an accurate manner. Additional information can be included as long as it does not contradict any stated facts."
- Informativeness: "Rank the explanations from best to worst according solely to their informativeness with respect to the facts in the given ground truth explanation. That is, any relevant factual information should be re-stated in an accurate manner and any necessary context should be provided."

Task-Specific Criteria:

- Urban Search and Rescue
  - Strategy: "Rank the explanations from best to worst according solely to how well the explanation identifies the behavior pattern of the agent given in the ground truth explanation. For example, does the agent focus on searching, rescuing, or some other behavior pattern."
  - Category: "Rank the explanations from best to worst according solely to how well the explanation identifies the type of the agent's goal given in the ground truth explanation. For example, is the agent heading towards a victim, piece of rubble, or unexplored room."
  - Goal: "Rank the explanations from best to worst according solely to how well the explanation identifies the location of the agent's goal given in the ground truth explanation. For example, does the explanation accurately identify the specific room the agent is heading to."
- Pacman
  - Goal: "Rank the explanations from best to worst according solely to how well the explanation identifies the agent's goal given in the ground truth explanation. For example, does the agent intend on eating food or a ghost."
- BabyAI
  - Rank the explanations from best to worst according solely to how well the explanation identifies the agent's goal given in the ground truth explanation. That is, does the explanation accurately identify that the agent is moving

towards the red ball, and if so does it accurately identify the position of the red ball."

## A.4 Environments

### A.4.1 Urban Search and Rescue

The Urban Search and Rescue (USAR) environment, as shown in Fig 6a, is a multi-agent cooperative Gridworld environment. Two heterogeneous agents, a medic and an engineer, are tasked with navigating an environment, removing rubble, and saving victims. Both agents can perform movement actions in which they can navigate to adjacent grid cells (left, right, top, bottom) as well as a unique role-specific action. The engineer can remove pieces of rubble while the medic can save victims. This environment is partially observable, and victims may be trapped under rubble and hidden. Thus, in order to save the victims, the agents must coordinate as the engineer must first remove rubble to locate victims, and the medic must then save them. In our environment, the grid contains 20 rooms arranged in a 4 × 5 grid with victims, rubble, and agents initialized in random locations. Our state representation contains the location of each agent, the location of any discovered rubble, the location of any discovered victims, and whether each room has been explored.

In Sec. 4.1 and Sec. 4.2, we leverage heuristic-based policies to sample states with known strategies and goals, which enables the synthetic generation of ground truth explanations. We analyze three unique behaviors:

- `Search`: The agents prioritize exploring all rooms before attempting to remove rubble and rescue victims.

- `Rescue`: The agents prioritize removing rubble and rescuing victims as they are discovered.

- `Fixed`: The agents ignore rubble, victims, and exploration and execute a fixed movement pattern. This behavior serves as a sanity check to examine the underlying assumptions of LLM-generated explanations.

### A.4.2 Pacman

The Pacman environment, as shown in Fig. 6b, is adapted from the classic arcade video game. In this environment, the agent is Pacman and must navigate a maze while consuming pellets to increase the score. The environment is fully observable and at every timestep Pacman must execute a movement action: move left, move right, move up, or move down. The environment contains 4 ghosts, and the episode ends with a negative reward if Pacman comes into contact with any of the ghosts. Our environment supports a mode shift in which Pacman is able to eat a power pellet which causes the ghosts to enter a "frightened" state for a short time. If Pacman collides with the ghosts in this state, they are consumed and the agent gains a positive reward. We leverage a custom state representation which indicates whether there is a wall above, below, to the left, or to the right of Pacman; whether there is a ghost above, below, to the left, or to the right of Pacman within a certain (3 cell) radius; the direction of the closest food pellet; and whether the ghosts are frightened.

### A.4.3 BabyAI

The BabyAI environment (Hui et al., 2020), as shown in Fig.6c, uses the `Gotolocal` scenario in which the agent must navigate to a red ball within a Gridworld environment filled with distractor objects. The environment is fully observable and the agent may choose to turn left, turn right, or move forward at every timestep. The state representation contains the position of each object relative to the agent's current position and orientation, as well as each object's type and color.

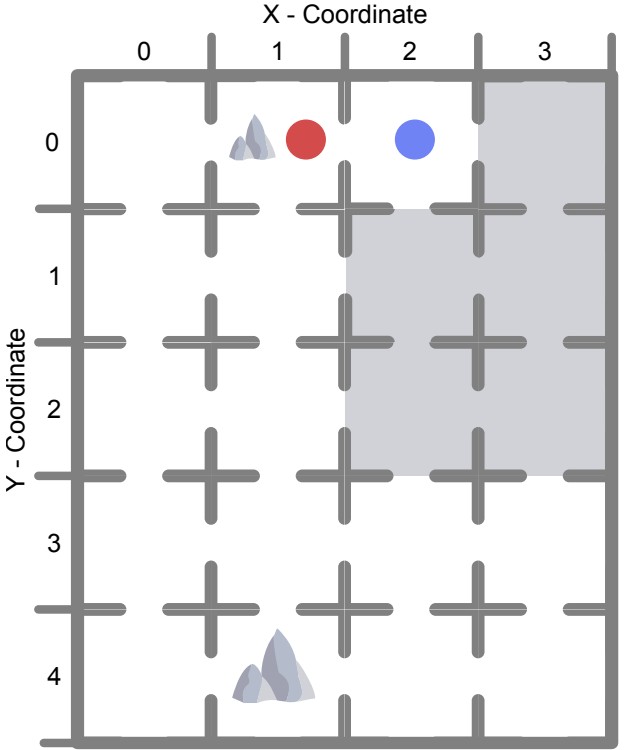

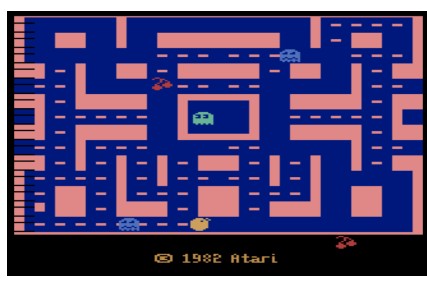

(b) Pacman environment, adapted from the Arcade Learning Environment and follows the same layout and rule set.

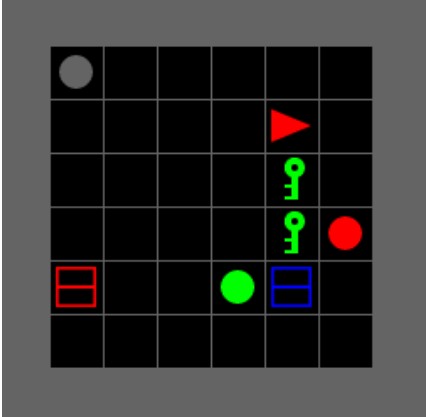

(a) Urban Search and Rescue environment. The red circle represents the medic, the blue circle represents the engineer, and the rocks represent rubble. Explored rooms have a white background, while unexplored rooms are gray.

(c) BabyAI environment. The red triangle represents the agent's position and orientation.

Figure 6: Visualizations of the three environments used in this work.

## A.5 Task-Specific Metrics

In this section, we provide details regarding the quantitative metrics we use in each environment for Sec. 4.1 and Sec. 4.2.

### A.5.1 Urban Search and Rescue

Suppose that the engineer is taking a `Rescue` strategy, and the decision path largely consists of decision rules operating over rubble locations.

- Strategy: Whether the overall strategy is identified. In this case, whether the explanation contains "The agent seems to prioritize the removal of rubble" for a `Rescue` engineer, as opposed to "The agent seems to be prioritizing searching" for a `Search` engineer.

- Category: Whether the agent's goal category was identified in the explanation. For example, if the explanation is "The engineer is prioritizing the removal of rubble and moves south to room (0,1) as that brings it closer to room (0,4) which contains rubble", then the explanation correctly identifies that the agent is seeking out rubble (which happens to be in room (0,4) in this case, although the specific location is not part of the category).

- Goal: Whether the agent's specific goal was correctly identified in the explanation. In the explanation above, if the agent's actual goal was to remove the rubble in room (0,4) (which we can observe by looking at future actions), then the goal is correct. However, if the agent instead moved south to room (0,1) in order to then move

east and remove rubble in room (1,1), the goal would be incorrect even though the category would still be correct.

### A.5.2 Pacman

- Goal: Whether the agent's specific goal was correctly identified in the explanation. Specifically, whether the agent is pursuing a pellet, super pellet, or a frightened ghost, or is the agent avoiding being eaten by a ghost.

### A.5.3 BabyAI

- Goal: Whether the agent's specific goal was correctly identified in the explanation. Specifically, whether the agent is pursuing the red ball (as opposed to a distractor), and if the location of the red ball been correctly identified.

### A.5.4 Action Prediction

All environments use the following agnostic action prediction metrics.

- Action: Whether the agent's next action was successfully predicted.
- Intent: Whether the agent's intent for taking the next action was successfully identified. If the explanation is "The engineer will move south to room (0, 3) to get closer to room (0, 4) which contains rubble." for a Rescue policy, Intent is correct. If the explanation is "The engineer will move south to room (0, 3) to get closer to room (1, 4) which is unexplored." for a Rescue policy (which should take rubble in room (0, 4) as priority), Intent is incorrect.

### A.6 Participant Study

This study evaluates three behavior explanation methods (Path, State, and Template) based on helpfulness, prediction accuracy, and hallucination detection. Each participant will complete two task types: comparison (Type A) and action prediction (Type B), using 8 cases split evenly. In Type A tasks, participants compare explanations pairwise, rank their helpfulness, and note if they seem hallucinated. In Type B tasks, participants predict the AI agent's next move, first without and then with explanations from each method, and note if they seem hallucinated. Maps and agent icons will be altered to avoid bias. Participants will provide demographic data, receive an introduction to the grid world environment, and complete tasks in the form of multiple choice questions. For each multiple-choice question, visualizations in GIF format that show the agent's trajectory will be provided. Results will be analyzed using statistical analysis to determine the most effective explanation method.

We use Qualtrics in participant demographics information and a locally operated website interface to collect user response. Regarding the study questions, our research survey consists of 21 questions of two categories: preference selection and action prediction. The survey involves participants viewing GIF figures to answer the questions, which makes it unsuitable for conversion into a static PDF format. The question order will be randomized according to our study design.

### A.6.1 Statistical Results

Here we describe our statistical approaches and report all relevant statistical values for our hypothesis in Sec. 5 to ensure transparency and reproducibility.

A binary logistic regression with controls is performed to test **H1.1**, **H1.2**, **H1.3**, **H3.1**, and **H3.2**. A McNemar's Test is performed to test **H1.4**.

- **H1.1 (Path >No Explanation):** ($coef = +2.650, SE = 0.436, oddsratio = 14.15, p < 0.001$).

- **H1.2 (Path >State):** ($coef = +2.898, SE = 0.617, oddsratio = 18.14, p < 0.001$).

- **H1.3 (Path >Template):** ($coef = +1.390, SE = 0.508, oddsratio = 4.014, p = 0.006$).

- **H1.4 (Path = Human):** ($p = 0.455$).

- **H3.1 (Subjective Hallucinations):**
    - Path vs. Human: ($coef = -1.713, SE = 0.447, OR = 0.18, p < .001$)
    - Path vs. No Explanation: ($coef = -0.395, SE = 0.340, OR = 0.67, p = 0.245$)
    - Path vs. State: ($coef = -0.918, SE = 0.519, OR = 0.40, p = 0.077$)
    - Path vs. Template: ($coef = -0.692, SE = 0.478, OR = 0.50, p = 0.148$)

- **H3.2 (Objective Hallucinations):**
    - Path vs. No Explanation: ($coef = +0.031, SE = 0.391, OR = 1.03, p = 0.936$)
    - Path vs. State: ($coef = -0.498, SE = 0.522, OR = 0.61, p = 0.340$)
    - Path vs. Template: ($coef = +0.451, SE = 0.648, OR = 1.57, p = 0.486$)
    - Path vs. Human: ($coef = +0.234, SE = 0.664, OR = 1.26, p = 0.724$)

We run one-sided signed Wilcoxon tests for **H2.1** and **H2.2**, and a two-sided Wilcoxon test for **H2.3**. In addition, we perform an ordinal logistic regression with controls to test **H4.1**, and **H4.2**.

- **H2.1 (Path >State):** ($W = 1778.00, n = 81, r = +0.061, p = 0.284$)

- **H2.2 (Path >Template):** ($W = 3016.00, n = 81, r = +0.709, p < 0.001$)

- **H2.3 (Path = Human):** ($W = 741.00, n = 74, r = -0.405, p < 0.001$)

- **H4.1 (Subjective Hallucinations):**
    - Path vs. Template: Path ($coef = -1.763, SE = 0.600, OR = 0.172, p = 0.003$), Template ($coef = -1.758, SE = 0.539, OR = 0.172, p = 0.001$)
    - Path vs. State: Path ($coef = +0.390, SE = 0.459, OR = 1.48, p = 0.396$), State ($coef = +0.657, SE = 0.465, OR = 1.93, p = 0.158$)
    - Path vs. Human: Path ($coef = +0.662, SE = 0.440, OR = 1.94, p = 0.133$), Human ($coef = -0.350, SE = 0.432, OR = 0.71, p = 0.417$)

- **H4.2 (Objective Hallucinations):**
    - Path vs. Template: Path ($coef = -1.049, SE = 0.503, OR = 0.35, p = 0.037$); Template has no hallucinations.
    - Path vs. State: Path ($coef = +0.515, SE = 0.468, OR = 1.67, p = 0.272$), State ($coef = +0.387, SE = 0.478, OR = 1.47, p = 0.418$)
    - Path vs. Human: Path ($coef = +0.096, SE = 0.429, OR = 1.10, p = 0.823$); Human has no hallucinations.

Furthermore, we found that certain demographic factors, specifically age and English proficiency, had a statistically significant effect on some of our hypotheses. However, since our participant population consists primarily of college students, as discussed in Sec. A.1, we refrain from drawing strong conclusions regarding these factors.

**Additional Statistics** We perform a binary logistic regression with controls to test whether subjectively perceived hallucinations are significantly correlated with objectively present hallucination, and find that the hypothesis is rejected $coef = +0.482, SE = 0.942, oddsratio = 1.620, p = 0.609$.

### A.6.2  *Instructions Given To Participants*

For our IRB-approved participant studies, we create a website containing multiple choice problems for the participants to select. Prior to answering the problems, they are required to go through a slide deck where textual instructions and a tutorial is given. The tutorial is to help the participants understand our interface and ensure that they understand the mechanisms of the environment.

The slide deck explains the roles and capabilities of the agents and basic knowledge about the USAR environment. Participants are then shown how natural language explanations are provided for each agent action and are instructed on how to assess these explanations for accuracy and helpfulness. The tutorial includes example scenarios and walks participants through two types of evaluation tasks: preference and next-action prediction.

### A.7  Distillation Performance Analysis

Our framework depends on a decision tree that is reflective of the agent's behavior, which requires the tree to have reasonable performance. We quantified the performance of the distilled policy by rolling out 5000 episodes with random initializations, and computing the action accuracy using the original policy's action as the target.

We note that our system only generates explanations when the action of the distilled decision tree matches that of the original policy, so as to avoid incorrect explanations.

|          | Search  | Rescue  | Fixed    |
|----------|---------|---------|----------|
| Medic    | 87.72%  | 87.05%  | 100.00%  |
| Engineer | 82.94%  | 96.42%  | 100.00%  |

Table 6: Accuracy of distilled decision tree policies for each behavior in the USAR environment.

|          | Pacman  | BabyAI  |
|----------|---------|---------|
| Accuracy | 77.90%  | 75.02%  |

Table 7: Accuracy of distilled decision tree policies for Pacman and BabyAI environments.

### A.8  Prompts

To ensure the reproducibility of this study, the prompts used for querying the large language model are provided. These prompts contain four segments: a) a concise description of the environment the agent is operating in, e.g., state and action descriptions, b) a description of what information the behavior representation conveys, c) in-context learning examples, and d) the behavior representation and action that we wish to explain. While the final part is produced by the surrogate model on a case-by-case basis, the first three parts are supplied for reference.

**Environment Description** provides the game rules and domain knowledge essential for the Large Language Model (LLM) to comprehend the situation. We also address potential contradiction between LLM knowledge and domain knowledge in this section.

**Behavior Representation Description** is provided to the LLM such that it knows how to reason with the information we provide. We emphasize that the LLM should *explain the policy of a neural network rather than formulating its own rationale*. This is because, as pointed out in Sec. 4.2, LLMs are prone to make assumptions over what the behavior should be as opposed to what the agent is actually doing.

**In-Context Learning (ICL) Examples** supply the LLM with instances of the expected outputs when an observation and action pair is presented. In this segment, we provide an example written by a human domain expert. Without prior knowledge of the policy's type, the domain expert is given the same description as above and produces an explanation based on the provided information. We incorporated three samples in each domain.

### A.8.1 Domain 1: Urban Search and Rescue

**Environment Description**

> The game is a search and rescue game set in a grid world. There are two agents: a medic and an engineer. The grid is made up of rooms, each represented by coordinates (x,y), where x represents the east-west direction, and y represents the north-south direction. Specifically, a larger x value corresponds to a location further east, and a larger y value corresponds to a location further south, with y=0 being the northernmost row and increasing y values moving southward.
>
> - Both the engineer and the medic can move to an adjacent room in any of the four cardinal directions (north, south, east, west) during a single move.
>
> - The medic has the ability to rescue a victim during a single move; however, this action cannot be performed concurrently with movement to an adjacent room.
>
> - The engineer has the ability to remove rubble during a single move; this action also cannot be performed concurrently with movement to an adjacent room.
>
> - Both agents can only perceive what's inside their current grid but have memory and can communicate instantly. Every grid visited by either of the agents is visible to both.
>
> - Rooms may initially be unexplored, in which case victims or rubble are assumed to be non-existent. Once explored, details about victims and rubble will be updated.
>
> - Rubble may or may not hide a victim. If a room contains rubble, victim is assumed to be non-existent. Removing the rubble may expose a hidden victim.
>
> - Rubble in the room won't affect the movement of the agents.

Although GPT-3.5 and GPT-4 displays considerable spatial reasoning skills, their default north/south direction is opposite to our configurations. Therefore, we incorporate explicit clarification of the direction in this segment. We also emphasized the difference between general LLM knowledge and domain knowledge in this segment. For instance, while the LLM assumes that rubble block the movement of agents, it does not in our context.

**Behavior Representation Description**

For Path:

> Given an observation of the environment, the agent chooses to look at a subset of features in order to choose its action, which we denote as "Features". These features represent parts of the observation that are directly related to why the agent chose its action. Given this subset of features that the agent looks at, provide a concise explanation for why the agent chose the action that it did. Keep in mind that the agent may not always be making optimal decisions, so consider that possibility if there seems like no reasonable goal the agent could be trying to accomplish from the set of features.

For `State`:

> You task is to explain why a neural network chooses the medic to take an action based on the given observation. The model's behavior can be understood from the following collection of observation and action pairs, given in the format of binary arrays (observation) and text (action). Each observation array represents the status of a particular feature; a '1' indicates the presence of that feature in the room, while a '0' denotes its absence. The features from 1 to 5 correspond to the presence of the exploration status, victim, rubble, engineer, and medic in the room, respectively.
>
> For example: Room (0, 0): [1, 0, 0, 0, 0] Room (0, 1): [1, 1, 0, 0, 0] In this example, Room (0, 0) is explored. Room (0, 1) is explored and contains a victim. Other features are absent for Room (0, 0) and (0, 1).
>
> The observation and action pairs that describes the model's behavior is given as follows:
>
> [Observation & Action]
>
> [Observation & Action]
>
> ...

For `No BR`:

> You task is to explain why a neural network chooses the medic to take an action based on the given observation. Each observation array represents the status of a particular feature; a '1' indicates the presence of that feature in the room, while a '0' denotes its absence. The features from 1 to 5 correspond to the presence of the exploration status, victim, rubble, engineer, and medic in the room, respectively.
>
> For example: Room (0, 0): [1, 0, 0, 0, 0] Room (0, 1): [1, 1, 0, 0, 0] In this example, Room (0, 0) is explored. Room (0, 1) is explored and contains a victim. Other features are absent for Room (0, 0) and (0, 1).

We note that both the `State` and `No BR` behavior representations utilize a vector state representation. This is because, unlike the `Path` representation which identifies a relevant subset of the observation, they must map the entire observation to text which is prohibitively long given limited-size context windows. In order to improve space efficiency and allow for both in-context learning examples and multiple observed states, we opt for a vector-format state representation and give a description of this format to the LLM. In the USAR domain, for instance, full-text translations of the state observation are approximately 1200 tokens in length (for a single observation), while vector representations are 440 tokens, depending on the observation in question.

**In-Context Learning (ICL) examples**

We incorporated three examples, one each from the three possible behaviors `Search`, `Rescue`, `Fixed`. Below is the in-context learning example for the `Search` policy.

ICL Example for `Path`.

> Features:
> room (0,1) doesn't contain rubble.
> engineer is not in room (0, 1).
> room (3, 2) doesn't contain rubble.

engineer is not in room (3, 2).
room (3, 3) doesn't contain rubble.
engineer is not in room (3, 3).
room (3, 4) contains rubble.
engineer is not in room (3, 4).
engineer is not in room (1, 1).
engineer is not in room (3, 0).
room (1, 3) doesn't contain rubble.
engineer is not in room (1, 3).
engineer is not in room (0, 4).
room (0, 4) doesn't contain rubble.
room (3, 1) doesn't contain rubble.
engineer is not in room (3, 1).
room (2, 4) doesn't contain rubble.
engineer is not in room (2, 4).
medic is not in room (2, 4).
medic is not in room (2, 1).
engineer is not in room (1, 2).
room (2, 3) doesn't contain a victim.
medic is not in room (3, 2).
medic is not in room (3, 1).
room (2, 3) doesn't contain rubble.
engineer is not in room (2, 3).
room (1, 4) doesn't contain rubble.
engineer is not in room (1, 4).
room (0, 3) has been explored.
room (0, 3) doesn't contain a victim.
medic is not in room (1, 4).
room (2, 2) contains rubble.

Action taken by the engineer:

engineer moves east to room (1, 2).

Explanation:

The engineer moves east to room (1, 2) because the engineer is currently in room (0, 2) and room (2, 2) contains rubble. From the set of features it appears the engineer is primarily looking for rooms containing rubble so that it can remove them, and since room (2, 2) contains rubble the engineer may be moving east to reach room (2, 2) and remove the rubble.

ICL Example for State and No BR.

Observation:
Room (0, 0): [0, 0, 0, 0, 0]
Room (0, 1): [0, 0, 0, 0, 0]
Room (0, 2): [1, 0, 0, 0, 0]
Room (0, 3): [1, 0, 0, 0, 0]
Room (0, 4): [1, 0, 0, 0, 0]
Room (1, 0): [0, 0, 0, 0, 0]
Room (1, 1): [1, 0, 1, 0, 0]
Room (1, 2): [1, 0, 0, 0, 0]
Room (1, 3): [1, 0, 0, 0, 0]

Room (1, 4): [1, 1, 0, 0, 0]
Room (2, 0): [1, 0, 0, 0, 0]
Room (2, 1): [1, 0, 0, 0, 0]
Room (2, 2): [1, 0, 0, 1, 0]
Room (2, 3): [1, 0, 0, 0, 0]
Room (2, 4): [1, 0, 0, 0, 0]
Room (3, 0): [1, 0, 0, 0, 0]
Room (3, 1): [1, 0, 0, 0, 0]
Room (3, 2): [1, 0, 0, 0, 1]
Room (3, 3): [1, 0, 0, 0, 0]
Room (3, 4): [1, 0, 0, 0, 0]

Action taken by the engineer:

engineer goes north to room (2, 1).

Explanation:

The engineer goes north to room (2, 1) because the engineer is in room (2, 2) and room (1, 1) contains rubble. The engineer is likely following a strategy of moving towards the closest rubble.

### A.8.2 Domain 2: Pacman

**Environment Description & Behavior Representation Description**

For Path

As a helpful agent, your role is to explain Pacman's behavior. Your task involves providing a clear and concise explanation of why Pacman chose a specific action given the observations of the environment. These observations encompass crucial details, such as the positions of walls, food, and ghosts in the cardinal directions (North, South, East, West), along with the current state of the ghosts (whether they are frightened or not). It's important to remember that Pacman's decision-making process may not always be optimal and might overlook some environmental details.

The following examples provide the observed features, the action Pacman took, as well as an explanation as to why the action was chosen:

For State

As a helpful agent, your role is to explain Pacman's behavior. Your task involves providing a clear and concise explanation of why Pacman chose a specific action given the observations of the environment. These observations encompass crucial details, such as the positions of walls, food, and ghosts in the cardinal directions (North, South, East, West), along with the current state of the ghosts (whether they are frightened or not). It's important to remember that Pacman's decision-making process may not always be optimal and might overlook some environmental details.

The following examples provide the observed features and the action Pacman took. Please use these to try to understand Pacman's reasoning:

For No BR

As a helpful agent, your role is to explain Pacman's behavior. Your task involves providing a clear and concise explanation of why Pacman chose a specific action given the observations of the environment. These observations encompass crucial details, such as the positions of walls, food, and ghosts in the cardinal directions (North, South, East, West), along with the current state of the ghosts (whether they are frightened or not). It's important to remember that Pacman's decision-making process may not always be optimal and might overlook some environmental details.

The following examples provide the raw observation, the action Pacman took, as well as an explanation as to why the action was chosen:

**In-Context Learning (ICL) Examples**

ICL Example for Path.

Features:
There is no food to the north.
There is no food to the west.
There is food to the south.
There is no wall to the west.
There is a wall to the east.
There is no ghost to the south.
There is no wall to the north.
There is no ghost to the west.

Action:

Pacman moves south.

Explanation:

Pacman decided to move south because there is food in that direction and no ghost, creating a safe path to increase Pacman's score.

ICL Example for State & No BR.

Observation:
wall_north:False
wall_south:False
wall_east:True
wall_west:False
ghost_north:False
ghost_south:False
ghost_east:False
ghost_west:False
food_north:False
food_south:True
food_east:False
food_west:False
scared_ghost:True

Action:

Pacman moves south.

Explanation:

Pacman decided to move south because there is food in that direction and no ghost, creating a safe path to increase Pacman's score.

### A.8.3   Domain 3: BabyAI

**Environment Description & Behavior Representation Description**

For Path:

As a helpful agent, your role is to explain the behavior of an agent operating in a 6x6 grid world. Your task is to provide a clear and concise explanation of why the agent chooses a specific action given the observations of the environment. While you do not know the particular goal of the agent, you know that the agent is attempting to locate and navigate to a specific object in the 6x6 grid environment. Overall, there are five objects that vary by position, type, and color. I will provide you with a description of the environment that includes information about the relative distances of each object with respect to the agent whose behavior you need to explain. I will also tell you the type and color of these objects such that you can reference them. The relative distance to these objects is provided in terms of grid cells with two values: the number of grid cells in front or behind the agent, and the number of grid cells to the left or to the right of the agent. The observation that you will receive to reason over the agent's behavior is the same information the agent used to make its decision. This information is referred to as features. Please note that the agent can only turn 90 degrees to the left or right with respect to its current heading, or move forward.

The following examples provide the observed features, the action the agent took, as well as an explanation as to why the action was chosen:

For State

As a helpful agent, your role is to explain the behavior of an agent operating in a 6x6 grid world. Your task is to provide a clear and concise explanation of why the agent chooses a specific action given the observations of the environment. While you do not know the particular goal of the agent, you know that the agent is attempting to locate and navigate to a specific object in the 6x6 grid environment. Overall, there are five objects that vary by position, type, and color. I will provide you with a description of the environment that includes information about the relative distances of each object with respect to the agent whose behavior you need to explain. I will also tell you the type and color of these objects such that you can reference them. The relative distance to these objects is provided in terms of grid cells with two values: the number of grid cells in front or behind the agent, and the number of grid cells to the left or to the right of the agent. The observation that you will receive to reason over the agent's behavior is the same information the agent used to make its decision. This information is referred to as features. Please note that the agent can only turn 90 degrees to the left or right with respect to its current heading, or move forward.

The AI agent's behavior can be understood from the following collection of observation and action pairs, given in the format of binary arrays (observation) and text (action). Each object in the world is associated with a 4-element binary array [x, y, type, color]. +y is in front of agent, -y is behind agent. +x is to agent's left, -x is to agent's right. Type 0 to 2 correspond to key, ball, box. Color 0 to 5 correspond to red, green, blue, purple, yellow, grey.

For example: Object 0: [1, 0, 1, 0] Object 1: [-2, 4, 2, 2] In this example, Object 0 is a red ball 1 grid cell to the agent's left. Object 1 is a blue box 2 grid cells to the agent's right and 4 grid cells in front of the agent.

Here is the collection of observation and action pairs. Please use these to try to understand the agent's reasoning.

[Observation & Action] [Observation & Action] ...

For No  BR

As a helpful agent, your role is to explain the behavior of an agent operating in a 6x6 grid world. Your task is to provide a clear and concise explanation of why the agent chooses a specific action given the observations of the environment. While you do not know the particular goal of the agent, you know that the agent is attempting to locate and navigate to a specific object in the 6x6 grid environment. Overall, there are five objects that vary by position, type, and color. I will provide you with a description of the environment that includes information about the relative distances of each object with respect to the agent whose behavior you need to explain. I will also tell you the type and color of these objects such that you can reference them. The relative distance to these objects is provided in terms of grid cells with two values: the number of grid cells in front or behind the agent, and the number of grid cells to the left or to the right of the agent. The observation that you will receive to reason over the agent's behavior is the same information the agent used to make its decision. This information is referred to as features. Please note that the agent can only turn 90 degrees to the left or right with respect to its current heading, or move forward.

The observation is given in the format of binary arrays. Each object in the world is associated with a 4-element binary array [x, y, type, color]. +y is in front of agent, -y is behind agent. +x is to agent's left, -x is to agent's right. Type 0 to 2 correspond to key, ball, box. Color 0 to 5 correspond to red, green, blue, purple, yellow, grey.

For example: Object 0: [1, 0, 1, 0] Object 1: [-2, 4, 2, 2] In this example, Object 0 is a red ball 1 grid cell to the agent's left. Object 1 is a blue box 2 grid cells to the agent's right and 4 grid cells in front of the agent.

The following examples provide the observed features, the action the agent took, as well as an explanation as to why the action was chosen:

As with the USAR domain, the we use a vector state format here in order to fit the ICL examples and state observations into the LLM context windows.

**In-Context Learning (ICL) Examples**

ICL Example for `Path`.

---

Features:
Object 4 is 3 grid cells behind.
Object 4 is 3 grid cells right.
Object 1 is 3 grid cells behind.
Object 3 is 1 grid cell in front.
Object 3 is green.
Object 4 is green.
Object 1 is purple.
Object 0 is 2 grid cells behind.
Object 2 is 1 grid cell in front.
Object 2 is red.
Object 2 is a ball.
Object 2 is a ball.
Object 2 is 1 grid cell right.
Object 3 is 3 grid cells right.
Object 4 is 3 grid cells right.
Object 2 is 1 grid cell right.

Action:

The agent moves forward.

Explanation:

The agent moves forward as it is attempting to reach the red ball which is 1 grid cell in front and 1 grid cell right, and there is no immediate obstacle in the agent's path.

---

ICL Example for `State & No BR`.

---

Observation:
Object 0: [-1, -2, 1, 2]
Object 1: [-5, -3, 1, 3]
Object 2: [-1, 1, 1, 0]
Object 3: [-3, 1, 0, 1]
Object 4: [-3, -3, 0, 1]

Action:

The agent moves forward.

Explanation:

The agent moves forward as it is attempting to reach the red ball which is 1 grid cell in front and 1 grid cell right, and there is no immediate obstacle in the agent's path.

---

