# OpenReview forum: "Model-Agnostic Policy Explanations with Large Language Models"
_colmweb.org/COLM/2025/Conference — COLM 2025_

### Official Review · Reviewer_i7vC · 2025-05-05

**Rating:** 8
**Confidence:** 3
**Ethics Flag:** 1

**Summary:**

This paper proposed a model agnostic policy explanation framework for agents using LLMs. They treat the original model as blackbox and use the observations and actions of the agents to generate explanations of the policy network of the agent. First, they distilled a black box policy into a decision tree. This decision tree policy contains a set of decision tree rules that approximate the decision making policy of the agent. Given a state and action, this decision tree policy generate a decision path that explains the agents behaviour. This decision path is fed to a LLM as prompt with other informations to generate the explanations. Authors conduct experiments with four datasets with four LLMs. They use another LLM (GPT 4o) to evaluate the explanations generated by the models with their decision path and without it. GPT 4o ranked these explanations across five different criterion and showed that their proposed framework does better in explaining the agent behaviour across those criterion. Authors also study with human evaluators if these explanations are helpful or not in real life.

**Reasons To Accept:**

1. The proposed model agnostic explanation generation for agent behaviour seems novel to me. It is simple yet quite powerful framework.
2. Evaluation study of the explanations with LLM and human is very rich and nicely done.
3. Authors very clearly demonstrated the impact of the proposed method with their detailed analysis.

**Reasons To Reject:**

NA

---

> ### Author Response · Authors · 2025-06-01
> **Response to Reviewer i7vC**
>
> We sincerely thank Reviewer i7vC for their thoughtful and encouraging comments. We're delighted that you found our framework novel and appreciated both the automated evaluation using LLMs and the user-perceived evaluation through our human study, as well as the depth and clarity of our analysis. If you have any further suggestions or questions, feel free to let us know!

---

### Official Review · Reviewer_QBcL · 2025-05-06

**Rating:** 7
**Confidence:** 3
**Ethics Flag:** 1

**Summary:**

The paper proposes a three‑stage framework that delivers faithful, low‑hallucination natural‑language rationales for an agent’s behaviour without peeking inside the agent’s model: it first distils the black‑box policy into an interpretable decision tree via DAgger, then extracts the local decision path that governs a queried state, and finally feeds that compact rule set to a large language model to generate the explanation.

Across three domains (multi‑agent urban search‑and‑rescue, Pacman and BabyAI) and four LLM families (GPT3.5, GPT4, Llama2, Starling), path‑based prompts consistently outperform baselines that expose either raw trajectories or no behavioural representation, slashing hallucination rates while producing explanations ranked more informative and strategy‑revealing by both automated scorers and human annotators.  A 29‑participant user study shows that these explanations let humans predict the agent’s next move better, matching expert‑written explanations, yet users still struggle to spot hallucinations, underscoring the value of the framework’s built‑in safeguards.

**Questions To Authors:**

If the distilled tree fails to accurately imitate the black‑box policy, the generated rationales may misrepresent the agent’s true decision process, no ? How to avoid this?

**Reasons To Accept:**

Strengths

- The 3 parts of the framework are simple and well explained (I like figure 1).
- Path‑based prompts consistently cut hallucination rates compared with baselines that see raw states or no behaviour representation.
- Tested on three distinct domains (USAR, Pacman, BabyAI) and four LLM families (GPT4 and 3.5, Llama2, Starling), showing robustness across tasks and language models.
- Experiments with re‑phrased prompts indicate the method is not highly sensitive to prompt engineering tricks.
- Combines automated metrics, manual annotation and behavioural testing, lending credibility to the claimed gains in faithfulness and usefulness.

**Reasons To Reject:**

Weakness:

- The decision‑tree surrogate can’t handle raw‑pixel inputs or very high‑dimensional sensory data, limiting applicability to vision‑heavy or continuous‑control tasks.
- User study sample is narrow (29 students): results may not generalise to diverse user populations or real‑world operator settings (but I appreciate that it's a human experience because it's already interesting).
- The cost of collecting trajectories and training the surrogate tree is not quantified, which could be prohibitive in complex environments.
- There are a few too many appendixes, it would be nice to make them a little more readable. A.1, A.2, A.3 and A.4 are good, but the others should be smaller.
- Typo: In table 1, some rows are missing the best values in bold.

---

> ### Author Response · Authors · 2025-06-01
> **Response to Reviewer QBcL (Part 3)**
>
> **Imperfect Imitation Risks**
>
> Thank you for raising this important point. We agree, and would like to address it from two aspects:
>
> **1. If the distilled tree has poor imitation accuracy**
>
> While we are reasonably confident that decision trees (and other interpretable surrogate models, see above) can accurately model complex policies, we agree that modeling errors can occur. However, the distilled tree (and thus the explanation) does not need to be perfectly accurate in all states to be useful. Since distillation is supervised learning, model accuracy can be assessed in states of interest and used to determine whether explanations are shown to end-users: if the model exhibits low accuracy in a given state, we withhold the explanation. This filtering process can be guided by downstream task performance metrics.
>
>
> **2. If the distilled tree does not represent the agent’s underlying decision process**
>
> While we cannot provide theoretical guarantees of faithfulness to the underlying decision process, this is a limitation shared by any post-hoc explanation method, as discussed in A.1. In practice, humans ascribe mental models to explain the behavior of other humans/pets/robots without access to the true underlying rationale and rely on these models to plan their own actions. Our mental models may not be true to the underlying reasoning process, but if they capture an agent’s high-level behavior then they are still useful.

---

> > ### Comment · Reviewer_QBcL · 2025-06-05
> >
> > Thank you for answering all my questions. It's clearer now. I'm still in favor of accepting the paper.

---

> ### Author Response · Authors · 2025-06-01
> **Response to Reviewer QBcL (Part 2)**
>
> **Computational Time Analysis**
>
> The computational complexity for distilling a decision tree to obtain a behavior representation is:
>
> $O(N \cdot M \cdot (C_{\text{bb}} + C_{\text{dt}} + C_{\text{env}}) + N \cdot C_{\text{train}})$
>
> where $N$ is the number of DAgger iterations; $M$ is the total number of timesteps collected per iteration; $C_\text{{bb}}$ is the time to query the black box policy for one action; $C_{\text{dt}}$ is the time to query the distilled decision tree for one action; $C_{\text{env}}$ is the time to take one environment step; and $C_{\text{train}}$ is the time required to fit a decision tree over the aggregated dataset.
>
> Assuming we use CART to fit a decision tree (which we did in this work), the complexity is $O(N \cdot M \cdot D \cdot \log(N \cdot M))$ where $D$ is the input dimension, which results in quadratic complexity in the second term with respect to the number of DAgger iterations $N$. However, in practice the number of iterations is usually small, e.g. $N \lt 10$, and in our work the wall-clock time required to distill a decision tree was on the order of ~1 minute. As the environment complexity grows $C_{\text{env}}$ will dominate the other terms, however, it is still linear with respect to the number of environment samples and so we don’t anticipate distillation costs to be prohibitive.
>
> **Appendix Readability**
>
> In response to your concern about readability, we will improve the clarity of the appendix, specifically A.5,6,7,8, to better highlight the extent of our additional analysis. We will retain the statistical reporting in A.6.1 to ensure transparency and reproducibility but greatly simplify the writing. In A.6.3, We will replace the participant study slide deck with a concise textual description of the instructions provided to participants. We will also improve clarity in A.5 and reorganize A.8 to better display the prompts.
>
> **Table 1 Typo**
>
> We would like to clarify that in Table 1, we bold only the values that are statistically significantly better, not simply the largest numbers in each group.

---

> ### Author Response · Authors · 2025-06-01
> **Response to Reviewer QBcL (Part 1)**
>
> We thank Reviewer QBcL for their thoughtful feedback. We’re glad you found our framework to be clearly presented, effective at mitigating hallucinations, and robust across diverse prompts, domains, and language models. We also appreciate your recognition of the breadth and depth of our experimental validation, as well as your acknowledgment of the credibility behind our claims.
>
>
> **Limited applicability**
>
> Regarding scalability to complex tasks, prior works have shown decision trees (and variants such as differentiable decision trees and gradient boosted decision trees [1][2]) are surprisingly effective at modeling even complex policies. In addition to the single-agent and multi-agent settings we examine in this work, previous papers have shown strong performance in continuous [3][4][5] and high-dimensional [2][6] state spaces as well as continuous control tasks [10]. So we have reason to believe that our approach would continue to perform well in a wide variety of tasks (beyond what we have already examined in this work).
>
> We do agree that images and/or multimodal inputs present challenges to classical decision trees. However, as we noted in our conclusion and future work (Sec. 6), we conjecture that our method would readily accommodate other classes of interpretable models that are better suited for these types of inputs [7][8]. Alternatively, in contemporary explainable AI literature, it is a common practice to first extract semantic or latent features from high-dimensional inputs to increase the interpretability of a black box model [9]; this is a strategy that could be applied to our proposed behavior representations by first extracting low-dimensional interpretable features and using these to distill a decision tree.
>
> [1] Silva, A., Gombolay, M., Killian, T., Jimenez, I., & Son, S. (2020). Optimization Methods for Interpretable Differentiable Decision Trees Applied to Reinforcement Learning. In Proceedings of the 23rd International Conference on Artificial Intelligence and Statistics (pp. 1855–1865). PMLR.
>
> [2] Si, S., Zhang, H., Keerthi, S.S., Mahajan, D., Dhillon, I.S. &amp; Hsieh, C.. (2017). Gradient Boosted Decision Trees for High Dimensional Sparse Output. Proceedings of the 34th International Conference on Machine Learning
>
> [3] Brița, C. E., van der Linden, J. G. M., & Demirović, E. (2025). Optimal Classification Trees for Continuous Feature Data Using Dynamic Programming with Branch-and-Bound. Proceedings of the AAAI Conference on Artificial Intelligence, 39(11), 11131-11139.
>
> [4] Quinlan, J. R. (1996). Improved use of continuous attributes in C4.5. Journal of Artificial Intelligence Research, 4(1), 77–90.
>
> [5] Ma, S., Zhai, J. Big data decision tree for continuous-valued attributes based on unbalanced cut points. J Big Data 10, 135 (2023). https://doi.org/10.1186/s40537-023-00816-2
>
> [6] Liu, W., & Tsang, I. W. (2017). Making decision trees feasible in ultrahigh feature and label dimensions. Journal of Machine Learning Research, 18(81), 1–36.
>
> [7] Frosst, N., & Hinton, G. (2017). Distilling a neural network into a soft decision tree. arXiv preprint arXiv:1711.09784.
>
> [8] Kontschieder, P., Fiterau, M., Criminisi, A., & Bulo, S. R. (2015). Deep neural decision forests. In Proceedings of the IEEE international conference on computer vision (pp. 1467-1475).
>
> [9] Koh, P.W., Nguyen, T., Tang, Y.S., Mussmann, S., Pierson, E., Kim, B., Liang, P. (2020). Concept Bottleneck Models. Proceedings of the 37th International Conference on Machine Learning.
>
> [10] Paleja, R.R., Chen, L., Niu, Y., Silva, A., Li, Z., Zhang, S., Ritchie, C., Choi, S., Chang, K.C., Tseng, H.E., Wang, Y., Nageshrao, S.P., & Gombolay, M.C. (2023). Interpretable Reinforcement Learning for Robotics and Continuous Control. ArXiv, abs/2311.10041.
>
>
> **Participant Population**
>
> While we acknowledge that our participant pool is limited to college-aged individuals, this initial study has provided valuable insights that inform the design of more comprehensive follow-up studies. In our in-person user study, we observed that participants found it challenging to detect hallucinations and preferred shorter responses (see response to Reviewer d9mA). This preliminary insight will guide us in designing further online studies with a diverse participant base. We will aim to design tasks to ensure that participants remain attentive and engaged while controlling for shortcut heuristics such as explanation length, thereby facilitating the consideration of all possible responses equally.

---

### Official Review · Reviewer_d9mA · 2025-05-12

**Rating:** 6
**Confidence:** 3
**Ethics Flag:** 1

**Summary:**

This paper proposes a method for NL explanations based on observed states and actions. While this is not a novel concept, this general approach is model-agnostic, which may lead to a relatively long-lived influence of the paper. A few task/environments are considered (two from urban search and rescue, one from Pacman (the game), one from BabyAI) and a few aspects are evaluated (e.g., comprehensibility, accuracy, explanation quality, hallucinations).

**Questions To Authors:**

-	Why are some of your references incomplete (e.g., Bills et al (2023))?
-	Can more be said (or more rationale provided) regarding human preferences and how well they actually align with correctness, utility, etc?

**Reasons To Accept:**

-	Humans (and an apparently good number of them) were involved in this study, the absence of whom is often cited as a limitation in other studies of this general type.
-	The solution to distill a decision tree is simple and straightforward.
-	A good number of experiments are included and they cover a sufficient breadth of aspects

**Reasons To Reject:**

-	It’s hard to see how the approach of using DTs will generalize both to more complex tasks or to tasks that involve more streams of data (esp multimodal data). This reliance relates also to the claims of ‘model agnosticity’, since not all forms of model may be as easily distilled to a DT.

---

> ### Author Response · Authors · 2025-06-01
> **Response to Reviewer d9mA (Part 2)**
>
> **Generalizability Concern**
>
> Regarding scalability to complex tasks, prior works have shown decision trees (and variants such as differentiable decision trees and gradient boosted decision trees [1][2]) are surprisingly effective at modeling even complex policies. In addition to the single-agent and multi-agent settings we examine in this work, previous papers have shown strong performance in continuous [3][4][5] and high-dimensional [2][6] state spaces as well as continuous control tasks [10]. So we have reason to believe that our approach would continue to perform well in a wide variety of tasks (beyond what we have already examined in this work). As such, we claim that our approach is model agnostic with respect to policies that can be represented by surrogate decision trees, which given our own experiments and the works above is a rather large set of models.
>
> We do agree that images and/or multimodal inputs present challenges to classical decision trees. However, as we noted in our conclusion and future work (Sec. 6), we conjecture that our method would readily accommodate other classes of interpretable models that are better suited for these types of inputs [7][8]. Alternatively, in contemporary explainable AI literature, it is a common practice to first extract semantic or latent features from high-dimensional inputs to increase the interpretability of a black box model [9]; this is a strategy that could be applied to our proposed behavior representations by first extracting low-dimensional interpretable features and using these to distill a decision tree.
>
> [1] Silva, A., Gombolay, M., Killian, T., Jimenez, I., & Son, S. (2020). Optimization Methods for Interpretable Differentiable Decision Trees Applied to Reinforcement Learning. In Proceedings of the 23rd International Conference on Artificial Intelligence and Statistics (pp. 1855–1865). PMLR.
>
> [2] Si, S., Zhang, H., Keerthi, S.S., Mahajan, D., Dhillon, I.S. &amp; Hsieh, C.. (2017). Gradient Boosted Decision Trees for High Dimensional Sparse Output. Proceedings of the 34th International Conference on Machine Learning
>
> [3] Brița, C. E., van der Linden, J. G. M., & Demirović, E. (2025). Optimal Classification Trees for Continuous Feature Data Using Dynamic Programming with Branch-and-Bound. Proceedings of the AAAI Conference on Artificial Intelligence, 39(11), 11131-11139.
>
> [4] Quinlan, J. R. (1996). Improved use of continuous attributes in C4.5. Journal of Artificial Intelligence Research, 4(1), 77–90.
>
> [5] Ma, S., Zhai, J. Big data decision tree for continuous-valued attributes based on unbalanced cut points. J Big Data 10, 135 (2023). https://doi.org/10.1186/s40537-023-00816-2
>
> [6] Liu, W., & Tsang, I. W. (2017). Making decision trees feasible in ultrahigh feature and label dimensions. Journal of Machine Learning Research, 18(81), 1–36.
>
> [7] Frosst, N., & Hinton, G. (2017). Distilling a neural network into a soft decision tree. arXiv preprint arXiv:1711.09784.
>
> [8] Kontschieder, P., Fiterau, M., Criminisi, A., & Bulo, S. R. (2015). Deep neural decision forests. In Proceedings of the IEEE international conference on computer vision (pp. 1467-1475).
>
> [9] Koh, P.W., Nguyen, T., Tang, Y.S., Mussmann, S., Pierson, E., Kim, B., Liang, P. (2020). Concept Bottleneck Models. Proceedings of the 37th International Conference on Machine Learning.
>
> [10] Paleja, R.R., Chen, L., Niu, Y., Silva, A., Li, Z., Zhang, S., Ritchie, C., Choi, S., Chang, K.C., Tseng, H.E., Wang, Y., Nageshrao, S.P., & Gombolay, M.C. (2023). Interpretable Reinforcement Learning for Robotics and Continuous Control. ArXiv, abs/2311.10041.

---

> > ### Comment · Reviewer_d9mA · 2025-06-05
> > **Generalizability**
> >
> > Thank you for those references and for your response. While I don't necessarily disagree with your 'reasons to believe', it would be expected to see these claims validated through experiments or at least _how_ they could be so validated. The challenge is really with the relatively incomplete empirical aspect -- if you could expand on these the paper may be improved.

---

> > ### Author Response · Authors · 2025-06-11
> > **Response to Generalizability**
> >
> > We truly appreciate you taking the time to discuss our paper and provide feedback to us. We would like to respond to your concerns about task complexity, as we consider multimodality (and image inputs) to be better left to future work given the large amount of experiments already included in the current version of this paper, as identified in our limitations section.
> >
> > Pacman and BabyAI are widely used both within the machine learning and explainability communities to better understand goal-oriented behavior [1,2]. These environments support complex behavior, e.g. our black box policy in Pacman dynamically transitions between multiple behavior modes: one where Pacman evades ghosts and another where Pacman chases ghosts (after consuming a power pellet).
> >
> > USAR is a heterogeneous multi-agent setting with partial observability. Due to heterogeneous action spaces agents must cooperate to fulfill their team objectives, and each agent exhibits different behavior modes. Task success depends on causal dependencies and long-horizon cooperation, i.e., a medic cannot rescue a victim until an engineer has removed the covering rubble. We emphasize that this environment presents meaningful challenges in planning and coordination.
> >
> > While we have provided references showing that decision trees (and variants such as gradient boosted trees) can be applied to a wide variety of tasks, we do believe that the set of tasks in this paper contain meaningful complexity while also being amenable to analysis.
> >
> > **How to apply our method to other tasks and inputs**
> >
> > Initial experiments suggest that our method can extend naturally to images with minimal changes. In the visual domains ALE [4] and Franka Kitchen [5], we defined a set of semantically meaningful features and leverage pre-trained foundation models to map raw image inputs to these features. We can then apply our three-step pipeline unchanged: 1) Distill a decision tree (or variant) over trajectories sampled from the black box model by pairing extracted features with black-box model actions. 2) Given a state we would like to explain, extract the corresponding decision path (this can also be applied to variants such as boosted trees [3]). 3) Query the LLM for an explanation given the decision path and following the prompt guidelines we have established. While our experimentation has thus far not included multi-modal inputs, we expect the same pipeline to hold as long as we extract semantically meaningful features from each modality.
> >
> > [1] Chevalier-Boisvert, M., Bahdanau, D., Lahlou, S., Willems, L., Saharia, C., Nguyen, T. H., & Bengio, Y. BabyAI: A Platform to Study the Sample Efficiency of Grounded Language Learning. In International Conference on Learning Representations.
> >
> > [2] Iyer, R., Li, Y., Li, H., Lewis, M., Sundar, R., & Sycara, K. (2018, December). Transparency and explanation in deep reinforcement learning neural networks. In Proceedings of the 2018 AAAI/ACM Conference on AI, Ethics, and Society (pp. 144-150).
> >
> > [3] Hatwell, J., Gaber, M. M., & Azad, R. M. A. (2021). gbt-hips: Explaining the classifications of gradient boosted tree ensembles. Applied Sciences, 11(6), 2511.
> >
> > [4] Bellemare, M. G., Naddaf, Y., Veness, J., & Bowling, M. (2013). The Arcade Learning Environment: An evaluation platform for general agents. Journal of Artificial Intelligence Research, 47, 253–279. https://doi.org/10.1613/jair.3912.
> >
> > [5] Gupta, A., Kumar, V., Lynch, C., Levine, S., & Hausman, K. (2019). Relay policy learning: Solving long-horizon tasks via imitation and reinforcement learning. arXiv preprint arXiv:1910.11956. https://arxiv.org/abs/1910.11956

---

> ### Author Response · Authors · 2025-06-01
> **Response to Reviewer d9mA (Part 1)**
>
> We thank Reviewer d9mA for their thoughtful feedback. We’re glad that you acknowledged the model-agnostic nature of our framework and its potentially long-lasting impact on the community. We also appreciate your positive remarks on the straightforwardness of our distillation approach, and the breadth of our experiments across environments, evaluation aspects, and participant study.
>
> **Reference Formatting**
>
> Thank you for pointing this out. Some of the references are blog posts (e.g. the one in question) and the style file did not properly display the URLs. We will fix this.
>
> **Human Preference Alignment**
>
> Regarding how well human preferences align with correctness, we found that participants tend to be overly generous in accepting explanations that did not align with the current agent’s strategy (as discussed in Section 5.2), and instead appear to rely on other factors when making their decisions. To investigate what these factors might be, we conducted additional analysis using explanation length as a variable. We found that longer explanations were significantly less likely to be preferred (p < 0.001 without controls; marginally significant at p ≈ 0.053 with controls). This suggests that the relationship between preference and correctness may be confounded or even dominated by differences in explanation length. In practice, we observed that participants tended to select shorter responses, regardless of their fidelity.
>
> As for explanation utility, we note that presenting both explanations side by side made it difficult to isolate the attention participants gave to each. However, our analysis using total time spent on a question found no evidence that longer engagement led to better preference for non-hallucinated explanations (p = 0.43 without controls; p = 0.38 with controls), nor to more accurate hallucination detection (p = 0.74 without controls; p = 1.00 with controls). This supports our finding that participants were not able to reliably detect hallucinations (Sec. 5.3), and further motivates the need for developing methods that inherently produce correct explanations.
>
> The additional statistical analysis and relevant discussion will be added to the Appendix.

---

### Decision · Program_Chairs · 2025-07-08

**Decision:**

Accept

**Comment:**

This paper aims to explain agent's actions without accessing the policy/model weights. The main approach is to distill the policy into a decision tree. While the idea is not particularly novel, this is the first time it's applied to LLMs and the method is evaluated comprehensively with human studies. The main limitation (which is mentioned by multiple reviewers) is whether DT can be scaled to more complex problems. Such limitation is hard to avoid in interpretability work though. I recommend accepting the paper.